# Visuo-thermal congruency modulates the sense of body ownership

Laura Crucianelli [1 ✉] & H. Henrik Ehrsson[1]

Thermosensation has been redefined as an interoceptive modality that provides information about the homeostatic state of the body. However, the contribution of thermosensory signals to the sense of body ownership remains unclear. Across two rubber hand illusion (RHI) experiments (N = 73), we manipulated the visuo-thermal congruency between the felt and seen temperature, on the real and rubber hand respectively. We measured the subjectively experienced RHI, the perceived hand location and temperature of touch, and monitored skin temperature. We found that visuo-thermal incongruencies between the seen and felt touch reduced the subjective and behavioural RHI experience (Experiment 1). Visuo-thermal incongruencies also gave rise to a visuo-thermal illusion effect, but only when the rubber hand was placed in a plausible position (Experiment 2) and when considering individual differences in interoceptive sensibility. Thus, thermosensation contributes to the sense of body ownership by a mechanism of dynamic integration of visual and thermosensory signals.

[1] Department of Neuroscience, Karolinska Institutet, Stockholm, Sweden. ✉email: laura.crucianelli@ki.se

Our bodies have the remarkable ability to constantly monitor their own temperatures. Beyond breathing, the regulation of body temperature (i.e., thermoregulation) is one of the most vital concerns for many homeothermic animals, including humans[1]. Both breathing and thermoregulation contribute to the maintenance of homeostasis (i.e., the active process of maintaining a particular physiological parameter at a relatively constant value[2]). Conscious sensations of warmth or coolness can be produced by objects that touch the body surface and communicate a potential change in body temperature. Though we often project the temperature to objects, the temperature is ultimately a physiological condition of the body surface itself. However, little is known about the contribution of peripheral thermosensation to the awareness of our body as our own (i.e., sense of body ownership).

Both temperature and touch are mediated by the skin, which participates in thermoregulation by cooling the body or conserving heat. Furthermore, thermal cues are important for discriminatory touch, contributing to the detection of the tactile identity of an object[3–5]. The process of perceiving temperature generates an inseparable affective state (i.e., pleasantness or unpleasantness[6]), a feeling that signals its homeostatic role and is directly dependent on the body's needs[7,8]. The affective aspects of such feelings correspond to the motivations that are essential for behavioural thermoregulation and homeostasis, that is, survival, by driving us to seek or avoid certain temperatures[6]. Thermal information is conveyed from the skin via thin unmyelinated c-fibres and takes a separate pathway to the brain compared to that of discriminative touch that travels through the spinal cord (spinal lamina I) and thalamus (ventral medial posterior nucleus) and reaches the insular cortex; the insular cortex is an important cortical region for processing interoceptive signals (including information from visceral organs). These are some of the reasons temperature and the affective component of touch, among others, have been redefined as interoceptive modalities[6,9,10] and separated from exteroceptive sensations (e.g., ref. [11]), such as information about the external environment (e.g., visual stimuli or discriminatory touch) and proprioception, which provides sensations of the positions of limbs and body parts in space[12]. Interoception refers to the perception and representation of internal signals about the physiological status of the body[6]. Thermosensation both provides information about the thermal state of one's own body (interoception) and about the thermal properties of the environment (exteroception), and, therefore, it can be used as an attractive model system of skin-based interoception[13]. However, despite the tight link among thermosensation, interoception, and bodily awareness, there has been hardly any investigation of the relationship among these three aspects[10,13].

Experimental paradigms using the rubber hand illusion (RHI[14]) have documented the mechanisms of integrating visual, vestibular, proprioceptive, and tactile input to give rise to a multisensory representation of our body[15]. During the RHI, participants experience ownership of a fake hand, which is placed in front of them, and touched in synchrony with their own hand, which is out of view. This dynamic, multisensory integration process that combines vision, touch, and proprioception is considered to be at the core of the sense of body ownership. One of the fundamental principles or constraints underlying the RHI is visuotactile congruency; that is, felt and seen touch must be spatially[16], temporally[17] and anatomically[18] congruent for the illusion to successfully take place[15]. Previous studies have also shown that congruency, in terms of tactile qualities between the touch that participants feel on their own hand and the one they see touching the rubber hand, might promote the induction of the illusion, particularly when comparing rough/hard vs. smooth/soft

materials[19–21] (but see ref. [22]). The tactile congruency rule relates to the correspondences of visual and tactile properties of the object in question-based on its macrogeometric and microgeometric features, such as texture[21]. Moreover, visuotactile mismatches between seen and felt touch can give rise to discrepancies between visually driven expectations[23,24] of how an object should feel and the actual perception of that object on the skin; such incongruence between visually driven expectations of macro- and microgeometric properties and actual somatosensory signals about such tactile information significantly supresses the illusion[19,21,25]. However, to the best of our knowledge, the effect of visuo-thermal discrepancies in the RHI remains unexplored. Here, we aim to address the question of whether (interoceptive) visuo-thermal congruency can be considered another basic congruence rule that affects the RHI. If this is the case, a mismatch between the seen and felt thermosensory stimuli (e.g., the temperature of the object on the rubber hand vs. the temperature of the object on the real hand) would impair the RHI experience.

In terms of the specific relationship between body temperature and the sense of body ownership, it has been suggested that the RHI might be accompanied by a physiological response, such as reduced skin temperature and, vice versa, manipulation of body temperature (i.e., cooling of the hand) may ease the induction of the RHI (e.g., refs. [26–28]). However, such a link has been difficult to replicate, and the finding is controversial (e.g., refs. [9,29,30]). Furthermore, the studies conducted thus far have mainly focused on investigating the link between body ownership and possible changes in skin temperature (thermoregulation, e.g., ref. [26]) rather than thermosensation. Although highly linked to one another, thermoregulation and thermosensation are two distinct phenomena, and they both contribute to the maintenance of thermoneutrality. Thermoregulation mainly involves automatic processes, while thermosensation is related to the conscious perception of thermal stimuli via the skin (e.g., refs. [31–33]). Hence, the involvement of temperature signals in body ownership is unclear, and in particular, it is unclear how such signals are combined with other sensory signals via the process of multisensory integration to generate a coherent sense of bodily self. Understanding the relationship between thermosensation and bodily awareness is important because our bodies are constantly immersed in an environment that has its own physical and thermal characteristics. Furthermore, in our everyday lives, we usually interact with objects for which we have previous knowledge of their typical temperature and thermal qualities. For example, we do not need to touch an ice cube to know that it is cold or a cup of tea to know that it is warm. The mere vision of these objects provides us with an embodied experience of 'what it would feel like' to touch or be touched by those objects[34]. In this context, a previous study reported that this experience of top-down, observed temperature can give rise to a visuo-thermal phenomenon, sometimes referred to as thermal contagion, whereby the experience of just looking at another's hand experiencing cold, in particular, was associated with corresponding changes in cold perception on the participant's hand; remarkably, even changes in participants' own skin temperatures were described[35], although this latter finding has yet not been replicated to the best of our knowledge. However, the thermal contagion phenomenon is not fully understood and has not been investigated in relation to the sense of body ownership.

Accordingly, we aimed to investigate the contribution of thermosensory signals to the multisensory integration process underlying the RHI. Across two experiments, we investigated whether visual and tactile experiences of touch using objects with congruent or incongruent thermal qualities can modulate the experience of body ownership. Specifically, we examined whether congruent or incongruent thermal and visual information would

modulate the sense of body ownership as quantified by classic subjective (illusion questionnaires) and indirect objective (shift in hand position sense towards the rubber hand, i.e., proprioceptive drift) measures. Moreover, to investigate a possible visuo-thermal illusion effect, that is, that the visual impressions of the object touching the rubber hand would influence the perceived temperature of the object in incongruent conditions but only when participants experienced some degree of rubber hand illusion, not when it was eliminated, participants completed a thermal matching task[10]. In its original format, participants were stroked with a thermode attached to a thermal stimulator at a reference temperature. The task of the participants is to match the reference temperature of the touch when presented among an ascending or descending series of successive warmer (up to $+8\,°C$) or cooler (up to $-8\,°C$) temperatures in a staircase procedure. Here, participants completed the thermal matching task following the rubber hand illusion, whereby the reference temperature was always the temperature felt during the induction of the illusion, thus providing an objective measure of thermosensation (see "Methods" for more details).

In secondary analyses, we also measured potential changes in body temperature by monitoring the skin temperature of participants' hands to explore whether visuo-thermal mismatches between the felt and seen touch in the RHI would give rise to homeostatic thermal feedback of the kind noted in ref. [35]. Finally, to target interoceptive mechanisms, and in keeping with previous studies[36], we collected interoceptive sensibility data by means of a self-report questionnaire, namely, the Body Perception Questionnaire[37], to explore potential relationships with the outcome measures of the illusion.

Overall, our results showed that visuo-thermal congruency enhances the RHI compared to visuo-thermal incongruency. This conclusion was supported by significant congruency effects from both the illusion questionnaire and proprioceptive drift in Experiment 1; thus, visuo-thermal congruency may constitute an additional rule that influences the RHI. As expected, such a visuo-thermal congruency effect was eliminated when the rubber hand was placed in an anatomically implausible position blocking the embodiment of the fake hand (Experiment 2).

## Results

### Experiment 1

*Proprioceptive drift.* First, we investigated the main effect of congruency on proprioceptive drift, regardless of temperature. The results of the Wilcoxon signed ranks test revealed a significant main effect of congruency ($Z$ ($n = 40$) $= -2.60$, $P = 0.008$), with congruent stimulation leading to higher proprioceptive drift scores ($M = 1.59$, SD $= 1.50$) than incongruent stimulation ($M = 0.78$, SD $= 1.62$, see Fig. 1b and Supplementary Fig. 2). This finding suggests that our incongruent condition successfully reduced the illusion in line with questionnaire findings. Next, we investigated the main effect of temperature on proprioceptive drift, regardless of congruency. The results of the Wilcoxon signed ranks test revealed a non-significant main effect of temperature ($Z$ ($n = 40$) $= -0.16$, $P = 0.88$, mean cold $= 1.28$, SD $= 2.11$; mean warm $= 1.09$, SD $= 1.90$). The interaction between congruency and temperature was non-significant ($Z$ ($n = 40$) $= -0.17$, $P = 0.89$). For confirmation purposes, we performed the same analysis using a repeated-measures ANOVA and found the same pattern of results (main effect of congruency: $F(1, n = 40) = 6.545$, $P = 0.015$; no significant main effect of temperature: $F(1, n = 40) = 0.135$, $P = 0.715$, and non-significant interaction between congruency and temperature: $F(1, n = 40) = 2.343$, $P = 0.134$).

*Thermal matching task.* The results of the 2 (congruency) × 2 (temperature) × 2 (increasing vs. decreasing) repeated-measures

ANOVA revealed a non-significant main effect of congruency on the thermal matching task ($F(1, n = 40) = 0.02$, $P = 0.89$). However, there was a significant main effect of temperature ($F(1, n = 40) = 96.53$, $P < 0.01$), with warm temperatures ($M = 2.19$; SD $= 3.12$) leading to higher errors in the thermal matching task compared to cold temperatures ($M = -1.36$; SD $= 3.24$). There was also a significant main effect of the staircase procedure ($F$ ($1, n = 40$) $= 46.12$, $P < 0.01$), with participants reporting lower performance on the thermal matching task (or higher visuo-thermal illusion) in the decreasing conditions ($M = 1.69$; SD $= 3.41$) compared to that in the increasing ($M = -0.88$; SD $= 3.42$) conditions. The interaction between congruency and staircase was not significant ($F(1, n = 40) = 3.70$, $P = 0.06$). No other significant interaction was found (congruency × temperature, ($F(1, n = 40) = 1.31$, $P = 0.26$); temperature × staircase ($F(1, n = 40) = 0.76$, $P = 0.39$); congruency × temperature × staircase ($F(1, n = 40) = 0.16$, $P = 0.69$) (see Fig. 2 and Supplementary Fig. 3).

We also conducted an exploratory ANCOVA to check whether individual differences in interoceptive sensibility might modulate the visuo-thermal illusion phenomenon; therefore, BPQ scores were included as covariates in the 2 (congruency) × 2 (temperature) × 2 (increasing vs. decreasing) repeated-measures ANCOVA. This analysis revealed a significant main effect of congruency on the thermal matching task ($F(1, n = 40) = 6.47$, $P = 0.015$), suggesting higher errors in the thermal matching task in the incongruent conditions than in the congruent conditions (see Fig. 2). This difference was higher in people with lower scores on the BPQ. There was a significant main effect of temperature ($F(1, n = 40) = 25.20$, $P < 0.01$). There was a non-significant main effect of the staircase procedure ($F$ ($1, n = 40$) $= 2.42$, $P = 0.13$). No significant interaction was found (congruency × staircase ($F(1, n = 40) = 3.21$, $P = 0.08$); congruency × temperature, ($F(1, n = 40) = 0.72$, $P = 0.40$); temperature × staircase ($F(1, n = 40) = 0.43$, $P = 0.52$); congruency × temperature × staircase ($F(1, n = 40 = 01.01$, $P = 0.32$) see Fig. 2).

The non-significant results of skin temperature monitoring are fully reported in the Supplementary Materials (Supplementary Table 1).

*Rubber hand illusion questionnaire.* First, we investigated the main effect of congruency on the composite illusion score, regardless of temperature. The results of the Wilcoxon signed ranks test revealed a significant main effect of congruency ($Z$ ($n = 40$) $= -2.18$, $P = 0.03$, mean congruent $= 1.67$, SD $= 1.15$; mean incongruent $= 1.36$, SD $= 1.18$), which suggests that thermal congruency might play a role in the subjective experience of the RHI. Next, we investigated the main effect of temperature on subjective illusion, regardless of congruency. The results of the Wilcoxon signed-ranks test revealed a non-significant main effect of temperature ($Z$ ($n = 40$) $= -0.75$, $P = 0.46$, mean cold $= 1.47$, SD $= 1.15$; mean warm $= 1.55$, SD $= 1.12$, Fig. 1a). The interaction between temperature and congruency was non-significant ($Z$ ($n = 40$) $= -1.71$, $P = 0.87$). These results are in line with the proprioceptive drift results reported above (see Fig. 1b).

We then investigated the three illusion items separately. The Wilcoxon signed-ranks test revealed a non-significant main effect of congruency in the location of touch ($Z$ ($n = 40$) $= -0.20$, $P = 0.88$, mean congruent $= 2.31$, SD $= 1.16$; mean incongruent $= 2.33$, SD $= 1.02$). Importantly, there was a significant main effect of congruency on causality of touch ($Z$ ($n = 40$) $= -2.47$, $P = 0.01$, mean congruent $= 1.83$, SD $= 1.54$; mean incongruent $= 1.34$, SD $= 1.69$) and ownership ($Z$ ($n = 40$) $= -2.02$, $P = 0.04$, mean congruent $= 0.86$, SD $= 1.63$; mean incongruent $= 0.4$, SD $= 1.82$), demonstrating that it was these phenomenological experiences that drove the significant overall congruence effect on the illusion composite score described above. A non-significant main effect of temperature was found for location of touch ($Z$ ($n = 40$) $= -1.47$, $P = 0.15$, mean

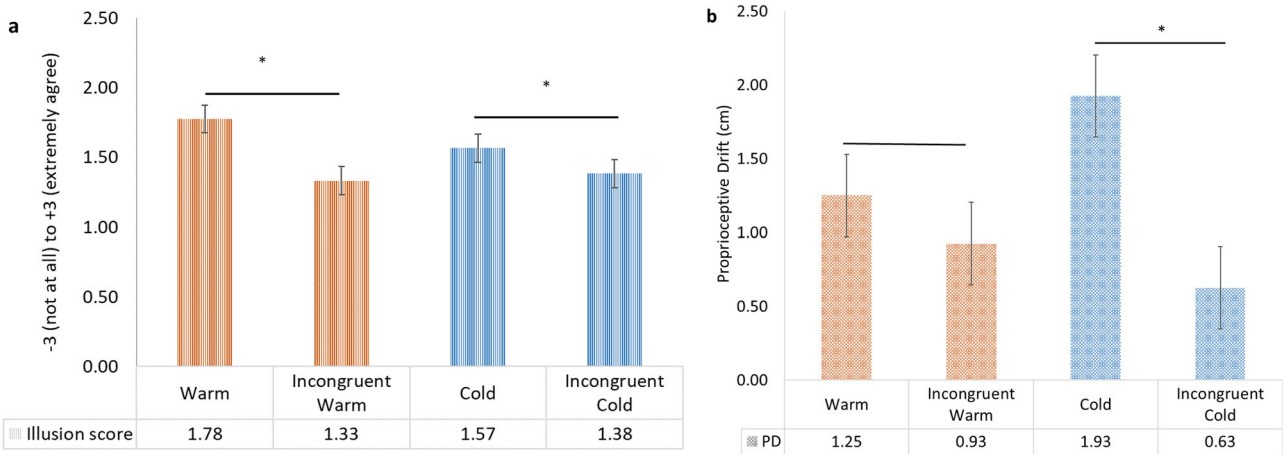

**Fig. 1 Illusion score and proprioceptive drift results for Experiment 1. a** Mean and standard errors for the composite illusion score (mean score of questionnaire Items 1–3. **b** Mean and standard errors for proprioceptive drift (PD). See Supplementary Fig. 1 and Supplementary Fig. 2 for figures with individual data points.

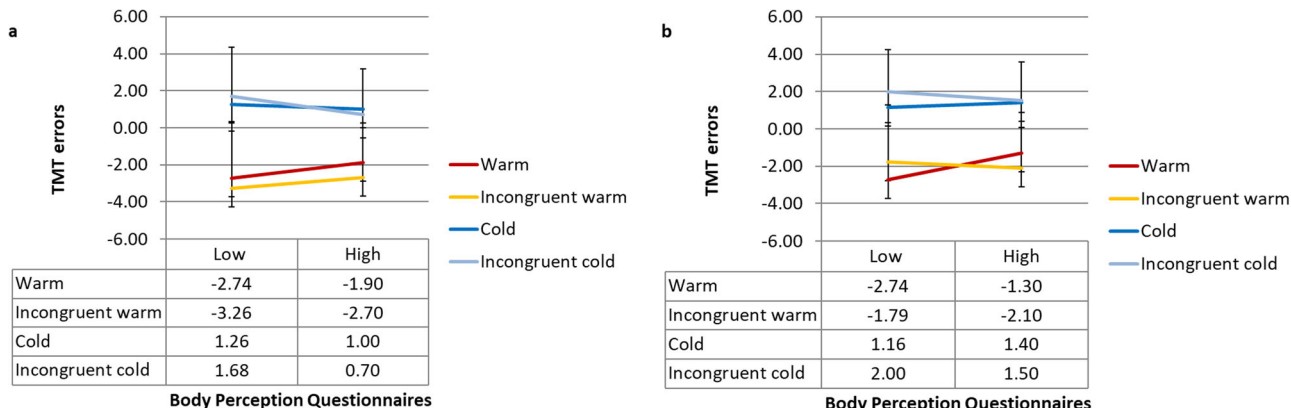

**Fig. 2 Visuo-thermal illusion effect following the illusion trials as a function of Body Perception Questionnaires (BPQ) scores for Experiment 1.** For the increasing (**a**) and the decreasing conditions (**b**). For visualisation purposes, we used a median split method to categorise BPQ outcomes into low and high score groups. Error bars denote ±1 standard error of the mean. See Supplementary Fig. 3 for figures with individual data points.

cold = 0.62, SD = 1.13; mean warm = 2.42, SD = 1.06), for causality of touch ($Z$ ($n = 40$) = −0.23, $P = 0.83$, mean cold = 1.56, SD = 1.64; mean warm = 1.61, SD = 1.46) and ownership ($Z$ ($n = 40$) = −0.66, $P = 0.52$, mean cold = 0.64, SD = 1.80; mean warm = 0.62, SD = 1.62).

In terms of the thermo-affective items of the questionnaire, Wilcoxon signed-ranks test results showed no significant main effect of congruency ($Z$ ($n = 40$) = −1.10, $P = 0.28$, mean congruent = 0.15, SD = 1.45; mean incongruent = 1.31, SD = 1.50) on pleasantness. However, there was a main effect of temperature ($Z$ ($n = 40$) = −3.10, $P < 0.01$) on the tactile pleasantness reported, with warm stimulation ($M = 1.82$, SD = 1.50) being rated as significantly more pleasant than cold stimulation ($M = 0.99$, SD = 1.65) regardless of congruency. For explicit cold perception, we found a non-significant main effect of congruency ($Z$ ($n = 40$) = −0.53, $P = 0.62$, mean congruent = −0.07, SD = 1.03; mean incongruent = 0.06, SD = 1.00). However, there was a significant main effect of temperature ($Z$ ($n = 40$) = −5.47, $P < 0.01$, mean cold = 2.33, SD = 1.00; mean warm = −2.33, SD = 1.20) on cold perception; that is, participants explicitly reported the temperature as being significantly colder when they perceived the cold temperature, regardless of congruency. Finally, we found a non-significant main effect of congruency on warm perception ($Z$ ($n = 40$) = −0.66, $P = 0.52$, mean congruent = 0.33, SD = 1.35;

mean incongruent = 0.19 SD = 1.19). However, there was a significant main effect of temperature on warm perception ($Z$ ($n = 40$) = −5.47, $P < 0.01$, mean warm condition = 2.45, SD = 1.64; mean cold condition = −1.94, SD = 1.37); that is, participants explicitly reported the temperature as being significantly warmer when they perceived the warm temperature compared to when they perceived the cold temperature, regardless of congruency. That is, no changes in cold or warm perception, as rated by the subjective questionnaires, were detected.

Finally, as an extra manipulation check and to control for confabulation, cognitive bias, and demand characteristics, we also averaged the scores of the six control items to obtain a composite control score and check for any effect of synchronicity, congruency, or temperature. The results of the Wilcoxon signed ranks test revealed a non-significant main effect of congruency ($Z$ ($n = 40$) = −1.00, $P = 0.32$, mean congruent = −0.98, SD = 1.07; mean incongruent = −1.07, SD = 1.15) or temperature ($Z$ ($n = 40$) = −0.82, $P = 0.42$, mean cold = −1.08, SD = 1.09; mean warm = −1.01, SD = 1.14) on the control score.

**Experiment 2**

*Proprioceptive drift.* As expected, we observed no RHI in the proprioceptive drift when comparing the synchronous and asynchronous conditions, as the rubber hand was always

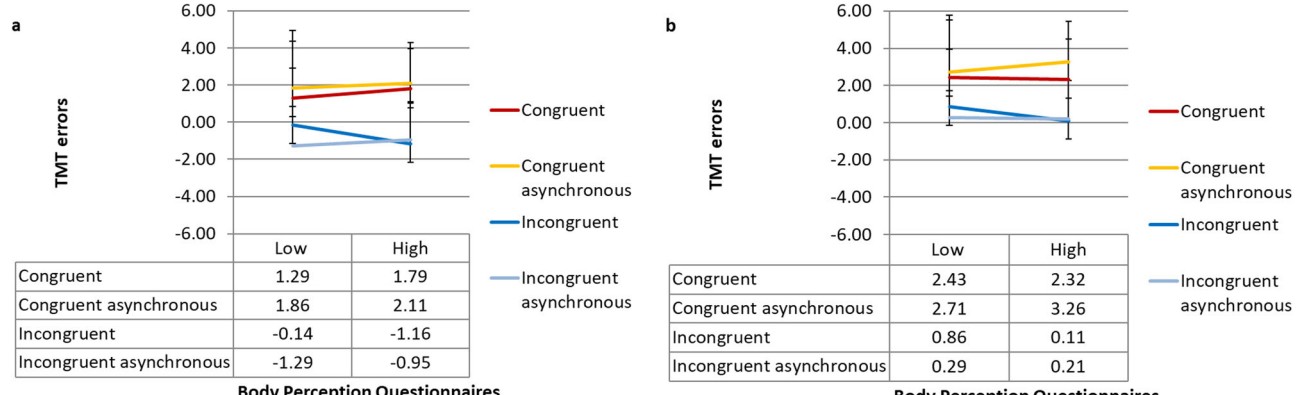

**Fig. 3 Visuo-thermal illusion effect following the illusion trials as a function of Body Perception Questionnaires (BPQ) scores for Experiment 2.** For the increasing (**a**) and the decreasing conditions (**b**). For visualisation purposes, we used a median split method to categorize BPQ scores into low and high groups. Error bars denote ±1 standard error of the mean. See Supplementary Fig. 4 for figures with individual data points.

presented in an anatomically implausible orientation. The results of the Wilcoxon signed-ranks test revealed a non-significant main effect of synchronicity ($Z$ ($n = 33$) = −0.89, $P = 0.38$, mean synchronous = −0.76, SD = 1.93; mean asynchronous = 0.05, SD = 2.75), suggesting that our experimental setup was successful in suppressing the embodiment of the rubber hand. To test whether our data provide evidence for the absence of an RHI effect, thus supporting the null hypothesis, we performed a Bayesian $t$ test. Bayesian analysis revealed a Bayes factor of 7.32 in favour of the null hypothesis of no RHI effect, indicating that the data were 7.32 times more likely under the null hypothesis than under the alternative hypothesis.

We performed an additional comparison between Experiment 1 and Experiment 2 to test the hypothesis of a weaker illusion in the latter due to anatomical incongruence; the results of the Mann–Whitney test revealed a significant main effect of the experiment on proprioceptive drift ($U$ ($n = 73$) = −4.00, $P < 0.01$), with participants in Experiment 1 showing higher values of proprioceptive drift of the left hand towards the rubber hand compared to that observed in Experiment 2 ($M_{Exp\ 1} = 1.92$, SD = 2.58; $M_{Exp\ 2} = −0.88$, SD = 3.26).

Next, we investigated the main effect of congruency on proprioceptive drift, regardless of synchronicity. The results of the Wilcoxon signed ranks test revealed a non-significant main effect of congruency ($Z$ ($n = 33$) = −0.75, $P = 0.47$, mean congruent = −0.72, SD = 1.51; mean incongruent = 0.00, SD = 2.97).

*Thermal matching task.* As in Experiment 1, we conducted a 2 (synchronicity) × 2 (congruency) × 2 (increasing vs. decreasing) repeated-measures ANCOVA with BPQ as a covariate. The results revealed a non-significant main effect of synchronicity on the thermal matching task ($F(1$, $n = 33$) = 0.72, $P = 0.40$) and a significant main effect of congruency ($F$ ($1$, $n = 33$) = 4.1, $P = 0.05$), with congruent stimulation ($M = 2.24$; SD = 2.01) leading to a higher visuo-thermal illusion effect (i.e., error in the thermal matching task) compared to incongruent stimulation ($M = −0.29$; SD = 2.07). However, participants reported errors closer to what they felt rather than to the seen temperature. There was a non-significant main effect of the staircase procedure ($F$ ($1$, $n = 33$) = 1.82, $P = 0.19$). No significant interactions were found between synchronicity and congruency ($F(1$, $n = 33$) = 3.62 $P = 0.07$), between synchronicity and staircase ($F(1$, $n = 33$) = 0.12 $P = 0.73$), between congruency and staircase ($F(1$, $n = 33$) = 0.03 $P = 0.85$), and among synchronicity × staircase ×

congruency ($F(1$, $n = 33$) = 0.63, $P = 0.43$) (see Fig. 3 and Supplementary Fig. 4). Thus, as we hypothesised, the visuo-thermal illusion effect was abolished when the rubber hand illusion was blocked by the anatomical incongruence manipulation.

The non-significant results of skin temperature monitoring are reported in the Supplementary Materials (see Supplementary Table 2).

*Rubber hand illusion questionnaire.* As expected, we observed no RHI in the questionnaire data when comparing the synchronous and asynchronous conditions, as the rubber hand was always presented in an anatomically implausible orientation. The results of the Wilcoxon signed-ranks test revealed a non-significant main effect of synchronicity on the illusion composite score ($Z$ ($n = 33$) = −1.74, $P = 0.08$, mean synchronous = −0.03, SD = 1.13; mean asynchronous = −0.32, SD = 1.11), confirming that our experimental manipulation of rubber hand orientation was successful in suppressing the rubber hand illusion (in line with previous studies[38–40]). To test whether our data provide evidence for the absence of a RHI effect (i.e., no significant differences between the synchronous and asynchronous conditions), thus supporting the null hypothesis, we performed a Bayesian $t$ test. Bayesian analysis revealed a Bayes factor of 1.23 in favour of the null hypothesis of no RHI effect, indicating that the data were 1.23 times more likely to be observed under the null hypothesis than under the alternative hypothesis. By convention, BFs between 0.33 and 3 are considered inconclusive[41,42]. Thus, the rubber hand illusion was denied by most participants in both synchronous and asynchronous conditions (mean negative ratings < −1 in all conditions, see Supplementary Fig. 1). These results are in line with the proprioceptive drift results reported above.

We performed an additional comparison between the rubber hand illusion in Experiments 1 and 2; the results of the Mann–Whitney test revealed a significant main effect of the experiment on the composite illusion scores ($U$ ($n = 73$) = −4.15, $P < 0.01$), with participants in Experiment 1 showing a higher level of subjective embodiment towards the rubber hand than participants in Experiment 2 ($M_{Exp\ 1} = 1.57$, SD = 1.27; $M_{Exp\ 2} = 0.15$, SD = 1.39).

Next, we investigated the main effect of visuo-thermal congruency on subjective illusion ratings (which were relatively low), regardless of synchronicity. The results of the Wilcoxon signed ranks test revealed a non-significant main effect of congruency ($Z$ ($n = 33$) = −0.56, $P = 0.58$, mean congruent = −0.15, SD = 1.15; mean incongruent = −0.21, SD = 1.08), which

suggests that the effect of thermal congruency was absent when the RHI was absent due to the manipulation of anatomical plausibility.

No significant results were found when investigating the three illusion items separately (all $Z$ values between $-1.95$ and $-0.22$, all $P$ values between 0.06 and 0.84). Moreover, a direct comparison with the data from Experiment 1 showed that referral of touch and ownership was rated significantly lower in Experiment 2 ($P < 0.05$; see Supplementary Materials for details). In terms of the thermo-affective items of the questionnaire, the results were in line with those of Experiment 1 (please see Supplementary Materials for further details).

Finally, as a manipulation check and to control for confabulation, cognitive bias, and demand characteristics, we also averaged the scores of the six control items to obtain a composite control score and check for any effect of synchronicity, congruency, or temperature. The results of the Wilcoxon signed-ranks test revealed a non-significant main effect of congruency ($Z$ ($n = 33$) $= -0.70$, $P = 0.49$, mean congruent $= -1.77$, SD $= 1.71$; mean incongruent $= -1.72$, SD $= 1.73$) and synchronicity ($Z$ ($n = 33$) $= -0.49$, $P = 0.63$, mean synchronous $= -1.76$, SD $= 1.69$; mean asynchronous $= -1.73$, SD $= 1.76$) on the control score of the illusion. The non-significant results of the correlational analyses across illusion measures are reported in the Supplementary Materials.

## Discussion

Across two experiments, we investigated the contribution of visuo-thermal signals to the sense of body ownership and visuo-thermal illusion phenomena. Overall, our results showed that visuo-thermal congruency may constitute an additional rule that influences the RHI. In particular, thermosensory signals were integrated with congruent visual and proprioceptive signals from the arm and contributed to the illusion of rubber hand ownership (Experiment 1). In addition, as expected, such a visuo-thermal congruency effect was eliminated when the rubber hand was placed in an anatomically implausible position blocking the embodiment of the fake hand (Experiment 2).

The results of the thermal matching task for Experiment 1 suggest that a visuo-thermal illusion takes place following the visuo-thermal incongruency in RHI, and this effect is modulated by individual differences in interoceptive sensibility. That is, participants with lower interoceptive sensibility showed a stronger visuo-thermal bias in the direction of the seen temperature compared to participants with higher interoceptive sensibility scores. Experiment 2 aimed to shed some light on the mechanism underpinning this phenomenon by showing that the visuo-thermal illusion does not occur when we experimentally block the illusion ownership of the rubber hand by placing the rubber hand in an implausible position. Collectively, the results of Experiments 1 and 2 suggest a link between the visuo-thermal congruency effect in the rubber hand illusion and the visuo-thermal bias of the felt temperature of the stimulated skin. Thus, the feeling of body ownership might be necessary for the visuo-thermal illusion effect to occur. Overall, the current results are important because they reveal two-way interactions between thermosensation and the multisensory experience of bodily self.

There is an increasing understanding that the awareness of our own body arises from an integration of information coming from outside our body (i.e., exteroception, such as visual and auditory cues, e.g., ref. [43]) and signals coming from the inside (i.e., interoception, such as heartbeat and pleasure from affective touch) (e.g., refs. [9,44,45]). Thus, the body and brain are equipped with sophisticated mechanisms that allow us to continuously integrate sensory information—such as vision, audition, and touch—to give rise to the awareness of our body as our own. The present

findings provide insights into such multisensory integration mechanisms underlying the RHI by showing that thermosensory signals are also integrated with congruent visual and proprioceptive signals from the body part and contribute to the sense of body ownership. Taken together, our findings corroborate the idea that the RHI takes place under multisensory integration rules (reviewed in ref. [15]) and further suggest that visuo-thermal congruency also contributes to the way we become aware of our body as our own. This is important because the ability to combine interoceptive and exteroceptive sensory signals can play an important role not only for the way we become aware of our body as our own but also for the way we resolve sensory incongruencies or conflicts in relation to our own body.

In particular, we believe that our study advances the body of knowledge in the multisensory integration field by focusing on one aspect that has been neglected thus far, that is, the thermal properties of touch, which are known to concurrently activate interoceptive and exteroceptive pathways[7,13,46]. Indeed, most of the studies thus far have focused on spatial and temporal constraints of multisensory integration in relation to visual, cutaneous, and proprioceptive feedback (see ref. [15] for a review). Here, we focused on the thermal interoceptive congruency between seen and felt touch; in addition, we showed that a mismatch between felt and seen temperatures can significantly diminish the rubber hand illusion phenomenon. In line with recent probabilistic models of own-body perception and the rubber hand illusion, information related to visuo-thermal congruence thus contributes to the automatic perceptual decision to infer a common cause for the visual, somatosensory, thermal, and other interoceptive sensations all originating from the owned rubber hand[47–50]. From this perspective, the present finding is noteworthy because it suggests an extension of the multisensory models of body ownership to include thermosensory information and multisensory congruence beyond temporal and spatial visuotactile (exteroceptive–exteroceptive) correlations, instead emphasising matching object thermal and visual properties (interoceptive–exteroceptive).

Traditionally, the role of interoception in multisensory integration has been quantified and measured by means of cardiac signals, often registered offline, and therefore considered a trait characteristic (e.g., ref. [9]). Here, we applied an online manipulation of interoception by providing tactile thermal signals during the RHI and considered the interplay between this online manipulation of skin-based interoception and offline/trait aspects of interoception, as measured by the interoceptive sensibility questionnaire. Focusing on online interoception is crucial because the contributions of thermal signals should be investigated in combination with visual and tactile information to understand the role of these signals in body ownership. For example, Siedlecka and colleagues[51] investigated the effect of body ownership on the perception of thermal stimuli following embodiment of the rubber hand. The visual presentations of thermal information were realistic and hence likely to induce a specific expected thermal sensation in real life, for example, an anticipated coldness when observing an ice cube and expected warm sensation from seeing a hand warmer. Participants performed a thermal change judgement task immediately following the RHI induction phase, where tactile thermal stimuli (e.g., plastic cube) were placed on the real hand, while visual-thermal stimuli (e.g., ice cube) were always placed on the rubber hand independently. The results showed that the sight of a visually thermal object touching the body surface can affect our thermal judgement only following a successful induction of the rubber hand illusion. This study provided validity for the use of an ice cube as an effective cold stimulus[51], which was confirmed by our results. In addition to underscoring that body ownership influences visuo-thermal

integration for external objects in line with Siedlecka's observation, our study expands on these findings by showing that a fake ice cube was successful in inducing the RHI when the other multisensory integration conditions were met and demonstrating that visuo-thermal congruence boosted this illusion. Thus, visuo-thermal integration influences body ownership, and not only the other way around. Importantly, the results of Experiments 1 and 2 combined suggest that the mechanism underlying the visuo-thermal bias in the perception of the felt temperature during the RHI cannot be explained as the simple thermal contagion phenomenon previously described in the literature (e.g., ref. [35]). Indeed, the effect was absent in Experiment 2 when we eliminated the RHI (by the strong violation of visuo-proprioceptive congruence occurring when placing the rubber hand in an anatomically implausible position). Thus, the current effect is different and linked to the sense of body ownership rather than mere visual observation.

In this regard, the dissociation between the explicit and implicit experience of the visuo-thermal illusion deserves some attention. Our results suggest that participants are quite accurate in reporting felt touch as cold or warm according to what they actually felt when explicitly asked to do so via the thermo-affective items of the illusion questionnaire. The fact that the questionnaire ratings were matched across conditions suggests that participants developed little awareness about the experimental manipulations, which thus speaks against any effect of confabulation, cognitive bias, and demand characteristics on participants' performance in the thermal matching task. However, at the objective level, we observed a bias in the direction of the seen temperature in that participants reported the temperature as significantly warmer or cooler (if they saw a warm or cold object touching the rubber hand, respectively) compared to what they actually felt (Experiment 1). Thus, the thermal matching task seems to be able to capture implicit thermosensory changes following multisensory integration manipulations. Importantly, the results of Experiment 2 further support this view by showing that participants do not experience the visuo-thermal illusion in the direction of the seen temperature when they do not recognise the rubber hand as their own; instead, they show a visuo-thermal bias towards the temperature they actually feel. As such, our results provide further support for the idea that the thermal matching task might be able to tap into skin-mediated interoceptive processes[10].

Furthermore, we demonstrated an additional link between thermosensation and interoception by showing that the visuo-thermal illusion is modulated by individual differences in interoceptive sensibility, whereby individuals with lower scores on the Body Perception Questionnaire (i.e., lower interoceptive sensibility) showed higher levels of visuo-thermal bias related to body ownership, that is, higher errors in the thermal matching task. This result can be interpreted in light of recent views describing interoception as a multidimensional construct[36]. In particular, interoceptive sensibility taps into individual differences in abilities to detect, attend to, and think about a wide range of internal signals in everyday life, such as stomach functions, thirst and body temperature. The fact that our results in the visuo-thermal illusion are modulated by such individual differences might suggest that metacognitive beliefs about the perception of interoceptive signals might guide the way we perceive such signals in relation to the awareness of our body as our own. Future studies could investigate whether it is possible to observe the relationship between visuo-thermal congruency in the RHI and visuo-thermal illusion without considering these individual differences in beliefs of interoception; thus, this point should be mentioned as a potential limitation of this study. Along this line, future studies could also use the present paradigm in functional neuroimaging experiments to investigate the neural mechanisms underlying the visuo-thermal congruency effect and related visuo-thermal illusion phenomena.

We believe that our study has a stronger ecological validity than previous work. For example, a recent study has investigated the top-down modulation of temperature perception during the RHI, following the principle that blue is associated with cold and red is associated with warm (according to the hue-heat hypothesis[52]). The results showed that warm temperature stimuli led to higher RHI vividness and were rated warmer when the thermal and visual (blue or red) stimuli were presented synchronously rather than asynchronously. Here, we did not find any significant difference between warm and cold stimuli in the RHI. However, we showed that visuo-thermal incongruencies between the felt and seen touch can significantly reduce the experience of ownership over the rubber hand. Importantly, we showed that such visuo-thermal effects also apply when we use everyday objects to deliver tactile stimulation to the rubber hand instead of red and blue lights, thus embracing a more ecological approach compared to the previous studies[51].

Finally, our skin temperature monitoring results do not support the idea that the RHI is accompanied by a significant drop in the temperature of the real hand[26], in keeping with recent findings and critical analyses of this putative phenomenon (e.g.,[9,30] but see[27]) (see Supplementary Table 1). Moreover, these findings also suggest that visuo-thermal mismatches between the felt and seen touch in the RHI do not significantly change the skin temperature to indicate a physiological homeostatic thermal feedback response (as in[35]), at least not significantly with the set of stimuli used in the current study (see Supplementary Table 1). This result is also in line with the view that thermosensation and thermoregulation involve different mechanisms and that they can deviate.

To conclude, the present experiments provided further support to the idea that the integration of interoceptive and exteroceptive signals is fundamental to the sense of body ownership by showing that a mismatch between what participants see and feel in terms of temperature can disrupt the perception of one's own body. Thermoregulation is an evolutionary requirement for our survival, and it is not surprising that our sense of body ownership could be coupled to fundamental interoceptive mechanisms of thermosensation. To experience a coherent and unitary sense of self, what we feel and what we see on our skin should match at any given time, and any incongruency should be resolved. We believe that our findings might pave the way for a better understanding of disorders of body awareness associated with temperature dysregulation, such as body disownership following right-hemisphere stroke[53,54] and eating disorders[55].

## Methods

**Experiment 1: hypotheses**. Experiment 1 investigated the effect of visuo-thermal congruency compared to incongruency on the RHI by manipulating the temperature of the seen touch on the rubber hand and of the felt touch on the real hand. This experiment used a fully factorial, 2 (object seen: cold vs. warm) × 2 (object felt: cold vs. warm) repeated measure design, where we manipulated the temperature of the felt stimuli on the real hand (cold, 24 °C and warm, 40 °C, which are − and + 8 °C from the neutral temperature of 32 °C, respectively) and the congruency of the temperature of the observed touch on the rubber hand (cold/ice cube and warm/hand warmer, see Fig. 4). We hypothesised that we would observe a main effect of congruency on the outcome measures of the illusion, i.e., a congruent temperature led to a stronger RHI than an incongruent temperature, as measured by means of an illusion questionnaire and proprioceptive drift. In the thermal matching task, we hypothesised that we would observe errors in the direction of the seen touch in the incongruent conditions such that participants would report the perceived temperature to be cooler than the actual felt temperature if they observed the rubber hand being touched by the ice cube. Similarly, they would report the perceived temperature to be warmer than the actual felt temperature if they observed the rubber hand being touched by the hand warmer. In other words, we expected incongruent conditions to give rise to a visuo-thermal

| EXPERIMENT 1 (n = 40) | | | EXPERIMENT 2 (n = 33) | | |
|---|---|---|---|---|---|
| **Horizontal set-up** | **FELT** | **SEEN** | **90° set-up** | **FELT** | **SEEN** |
| Congruent — Cold | 24°C | | Congruent — Synchronous | 24°C | |
| Congruent — Warm | 40°C | | Congruent — Asynchronous | 24°C | |
| Incongruent — Cold | 24°C | | Incongruent — Synchronous | 32°C | |
| Incongruent — Warm | 40°C | | Incongruent — Asynchronous | 32°C | |

**Fig. 4 Summary of the experimental conditions across the two experiments.** The felt touch on the real hand was delivered at cold (24 °C), neutral (32 °C) or warm (40 °C) temperatures. The touch on the rubber hand was delivered with a fake ice cube or a hand warmer.

illusion, whereby participants would report the perceived temperature to be closer to the seen temperature. We assumed that this visuo-thermal illusion would be driven by illusory rubber hand ownership, as the fake hand was synchronously stroked in all conditions and placed in an anatomically congruent position, which should also lead to a certain degree of RHI in the visuo-thermal incongruent condition (an assumption tested in Experiment 2, see further below), in line with the earlier observation that visuotactile incongruencies tend to reduce rather than completely eliminate the illusion during synchronous stroking[12,21,25]. The relationship between interoceptive sensibility and the outcome measures of the illusion was also explored to investigate whether interoceptive sensibility would modulate the visuo-thermal illusion.

**Experiment 1: participants**. Forty participants (24 women, mean age = 26.67; SD = 4.72) were recruited using social media and advertising on the Karolinska Institutet campus. No outliers were identified (values above or below 2.5 SD from the mean in each variable). A priori power analysis (G*Power 3.1[56]) based on previous studies using a within-subjects design in the RHI (e.g.,[9,43]) suggested that the present sample provided enough power to detect our effects of interest (power 0.92; α = 0.05, effect size d = 0.6, two-tailed). Inclusion criteria were being 18–40 years old and being right-handed. Exclusion criteria were having a history of any psychiatric or neurological conditions, taking any medications, having sensory or health conditions that might result in skin conditions (e.g., psoriasis), and having any scars or tattoos on their left forearm or hand. The study was approved by the Swedish Ethical Review Authority. All participants provided signed informed written consent, and they received a cinema ticket as compensation for their time. The study was conducted in accordance with the provisions of the Declaration of Helsinki in 1975, as revised in 2008.

**Experiment 1: materials and experimental procedure**. Upon arrival and after signing the consent form, participants provided demographic information and completed the Body Perception Questionnaire (BPQ short version[37], Porges, 1993) as measures of interoceptive sensibility (mean BPQ = 29.36; SD = 10.62). The testing room was kept at a constant temperature measured by means of a room thermometer that was checked at the beginning and at the end of each testing session (mean_pre = 21.48 °C, SD = 0.35; mean_post = 21.76 °C, SD = 0.40).

Next, a RHI was performed using a custom-made panel (50 × 60 cm) to control visual feedback of the participants' arm own left arm and the rubber hand during the experiment (see Fig. 2). The distance between the real left hand and rubber hand was 15 cm. Before starting the experimental procedure, three thermal sensors (Biopac MP150) were placed on the participants' skin to continuously measure the temperature of the skin; one sensor was placed on the right dorsal hand, one was placed on the left dorsal hand and one was placed on the left dorsal forearm (see Fig. 5 for a clear view of the sensor positions). Two fake sensors were also placed on the left rubber hand and forearm to avoid any visual mismatch between the real and fake hand (see Fig. 5). The experimenter then sat opposite the participant and stroked the participant's own hand and the rubber hand for 1 min at a CT-optimal velocity of 3 cm/s (see refs. [9,45]), resulting in a total of 15 strokes, with a 1 s break between strokes. The thermal stimuli on the participant's hand were delivered by means of a thermode at either warm or cold temperatures (Somedic MSA Thermal Stimulator see Fig. 5a, as in ref. [52]).

Concurrently, participants looked at the rubber hand being stroked by two everyday objects, a (fake) ice cube (cold condition) and a transparent hand warmer (warm condition) (see Fig. 2a). These objects are usually associated with specific cold and warm temperatures but characterised by neutral colour to avoid any hue-heat effects[52]. Participants were visually familiarised with the objects before commencing the experimental procedure to ensure that all participants started with an (at least basic) equal knowledge of the objects. Experiment 1 consisted of 12 semi-randomised trials: first, 4 localisation trials (congruent and incongruent warm; congruent and incongruent cold) to measure proprioceptive drift; second, 4 trials followed by the thermal matching task to measure the perceived temperature during the RHI (see below); and finally, four trials to measure the subjective RHI (i.e., illusion questionnaire[14]). The conditions within each experimental block were counterbalanced and randomised for each participant to minimise possible order effects. Prior to commencing the next condition, participants had a resting period, during which they were instructed to freely move their left hand.

During the proprioceptive drift trials, participants were asked to close their eyes, and the experimenter positioned the participants' right index fingers on the right side of a metal ruler, which was placed on the table five centimetres over both the right hand and the rubber hand. The starting point on the metal ruler was varied randomly by the experimenter. The custom-made divider had a window in the middle that was open during the proprioceptive drift trials, allowing the participants to reach the left-hand side of the panel (see Fig. 5b, see ref. [57] for a similar procedure). Then, the participants moved their right index fingers to signal where they felt that their left index finger was located. The experimenter noted the value and calculated the difference in cm between the actual location of the index finger and the one indicated by the participants as a measure of localisation error. The same procedure was repeated before and after each stimulation period. The proprioceptive drift was then calculated as the difference between the localisation errors before and after the illusion trials.

The thermal matching task[10] was performed once before (i.e., baseline measure) and once after each visuotactile stroking condition as a measure of visuo-thermal illusion; that is, this task was performed to determine whether the seen temperature might have influenced the felt temperature perceived during the RHI at an implicit level. We followed the same procedure and used the same equipment as in ref. [10]. In the pre-trial (repeated only once at the beginning of the experimental session to familiarise participants with the procedure), participants were stroked with a 25 × 50 mm thermode attached to a thermal stimulator on the real hand (Somedic MSA, SenseLab, Sweden) at a reference neutral temperature of 32 °C. The task followed a staircase procedure, that is, the temperature was either increased (from 24 °C) or decreased (from 40 °C) towards the reference temperature in discrete steps of 2 °C. Participants were instructed to try to reach the reference temperature that they previously experienced by verbally saying 'stop' when the test temperature presented among warmer or cooler temperatures in the staircase procedure matched the reference temperature. In the post-trial thermal matching task, we repeated the same procedure, but participants were asked to match the temperature they had felt on their real hand while they were looking at the rubber hand (reference temperature) by verbally saying 'stop' when the series of warmer and cooler temperatures was presented (after the illusion induction stimulation). This procedure was repeated twice after each trial (one increasing and one decreasing staircase), and participants were always asked to refer to the temperature they

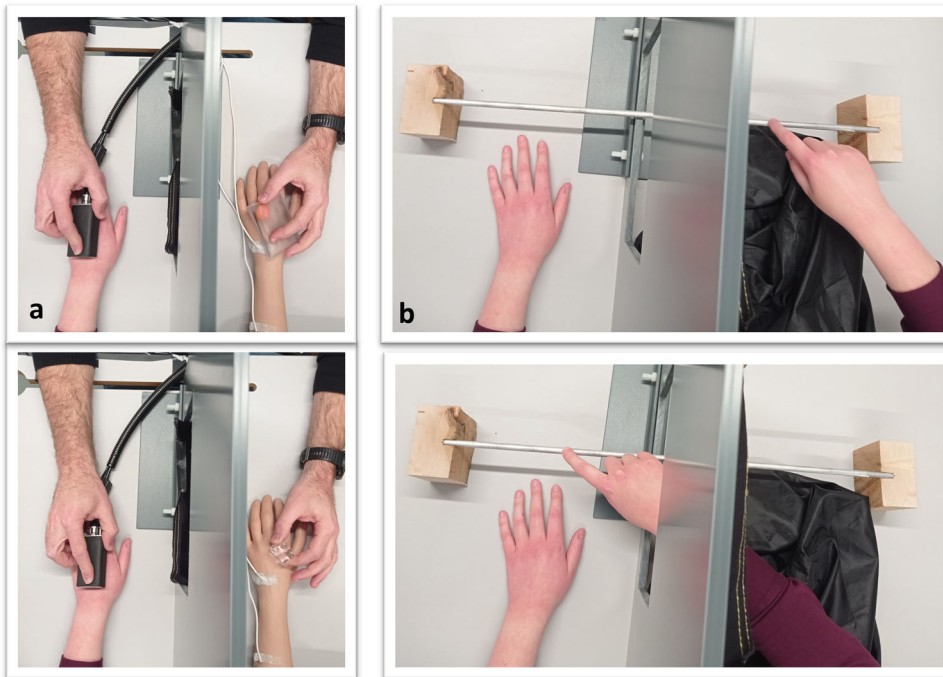

**Fig. 5 Experimental setup for Experiment 1. a** The thermode of the Somedic MSA Thermal Stimulator was used to deliver touch at warm (40 °C) or cold (24 °C) temperatures on the participant's left hand, while participants watched the rubber hand being touched by a hand warmer (top) or an ice cube (bottom). **b** Visual representation of the proprioceptive drift procedure.

**Table 1 Illusion questionnaire, with the classic illusion statements, the control, and thermo-affective items (specified in bold).**

| During the experiment there were times when... | Category |
|---|---|
| 1 It seemed as if I were feeling the touch of the object in the location where I saw the rubber hand touched | **Illusion statement (referral, location)** |
| 2 It seemed as though the touch I felt was caused by the object touching the rubber hand | **Illusion statement (referral, causality)** |
| 3 I felt as if the rubber hand was my hand | **Illusion statement (ownership)** |
| 4 It felt as if my (real) hand was drifting towards the rubber hand | **Control item** |
| 5 It seemed as if I might have more than one left hand or arm | **Control item** |
| 6 It seemed as if the touch I was feeling came from somewhere between my own hand and the rubber hand | **Control item** |
| 7 It felt as if my (real) hand was turning 'rubbery' | **Control item** |
| 8 It appeared (visually) as if the rubber hand was drifting towards my hand | **Control item** |
| 9 The rubber hand began to resemble my own (real) hand, in terms of shape, skin tone, freckles or some other visual features | **Control item** |
| 10 The sensation I felt on my arm and hand was cold | **Thermo-affective** |
| 11 The sensation I felt on my arm and hand was warm | **Thermo-affective** |
| 12 The sensation I felt on my arm and hand was pleasant | **Thermo-affective** |

Participants responded using a rating scale ranging from −3, completely disagree to +3, completely agree.

perceived on their left hand during visuotactile stimulation. Finally, participants completed the illusion questionnaire (see Table 1). Participants were presented with the items in a randomised order, and they were asked to respond verbally using a rating scale ranging from −3 (strongly disagree) to +3 (strongly agree).

**Experiment 1: experimental design and statistical analysis**. Data were analysed using SPSS 26. First, we tested for normality by means of Shapiro–Wilk tests, and data distribution was also checked by means of visual exploration (Q–Q plots). Proprioceptive drift was found to be non-normally distributed ($P$ values < 0.05), and given the ordinal nature of the questionnaire data, these data were analysed using non-parametric statistics by means of the Wilcoxon signed-ranks test. The limb temperature data and thermal matching task data were found to be normal, so repeated-measures ANOVAs were applied, followed by Bonferroni corrected post hoc analyses when appropriate. All the reported statical tests are two-tailed.

This experiment followed a 2 (congruency) × 2 (temperature) fully factorial repeated-measures design. The incongruent conditions were thus used as control conditions for the occurrence of the illusion. We focused on the effect of congruency and temperature on our illusion measures (i.e., illusion questionnaire, proprioceptive drift, and thermal matching task) by comparing results of the congruent conditions

with those of the incongruent conditions, in both warm and cold conditions, and by comparing the congruent warm and cold conditions to investigate the effect of temperature. To investigate the main effect of congruency, the two congruent and the two incongruent trials were averaged to obtain one congruent and one incongruent score; these scores were compared using the Wilcoxon signed-rank test. Similarly, to investigate the main effect of temperature, the two warm and two cold trials were averaged to obtain one warm and one cold score, and these were compared using the Wilcoxon signed-rank test. The interaction between congruency and temperature was analysed by calculating the difference between congruent and incongruent scores in the warm and cold conditions separately; subsequently, a Wilcoxon signed-rank test was performed to compare these two different scores.

The proprioceptive drift score was calculated as the difference between the pre-trial and post-trial finger localisation measures (in line with previous studies, e.g., refs. [9,58,59]). We also calculated the proprioceptive shift to run Spearman correlational analyses by computing the difference between the congruent and incongruent conditions for the cold and warm conditions separately.

For the thermal matching task, we calculated the error between the reference temperature and the matched temperature, and we compared the errors post-trial by means of a 2 (temperature) × 2 (congruency) repeated-measures ANOVA.

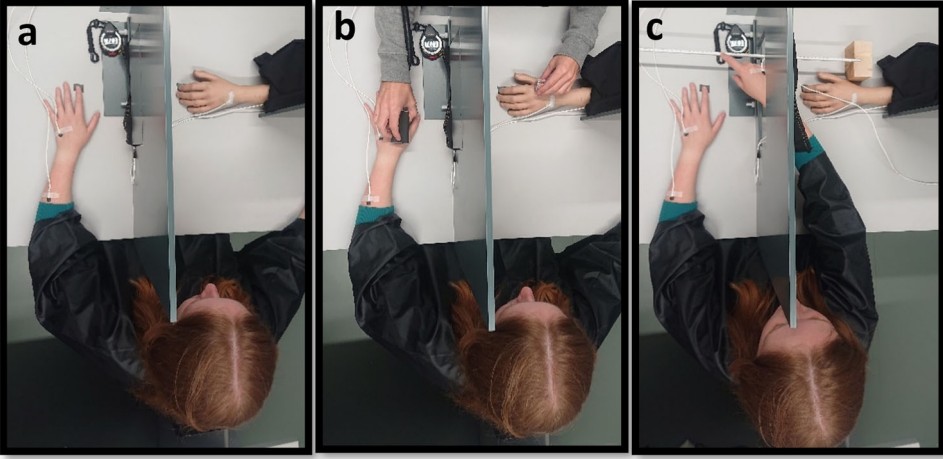

**Fig. 6 Experimental setup for Experiment 2. a** The two Biopac thermosensors on the left arm and hand are visible. The rubber hand had the same sensors to avoid any mismatch between the real and the rubber hand. **b** The thermode of the Somedic MSA Thermal Stimulator was used to deliver a touch at 24 °C or 32 °C on the participant's left hand, while participants watched the rubber hand being touched by an ice cube. **c** Visual representation of the proprioceptive drift procedure.

Similarly, for skin temperature, we performed a 2 (pre vs. post) × 3 (location: left arm vs. right hand vs. left hand) × 2 (congruency) × 2 (temperature) repeated-measures ANOVA (see Supplementary Table 1). We also calculated the thermal shift to run correlational analyses by computing the difference in performance at the thermal matching task between the congruent and the incongruent conditions for the cold and warm conditions separately.

For the illusion questionnaire, we focused on the items separately and calculated an illusion composite score (i.e., average of Items 1–3) and a control composite score (i.e., average of Items 4–9) (see refs. [45,60] for a similar approach). We also calculated an illusion shift score to run correlational analyses by computing the difference between the congruent and the incongruent condition for the cold and the warm conditions separately.

The illusion shift, proprioceptive shift, and thermal shift were then used in Spearman correlational analyses to investigate the relationship between the outcome measures of the RHI and the interoceptive sensibility measure. The results of these analyses are reported in the Supplementary Materials.

**Experiment 2: hypotheses**. The results of Experiment 1 showed a visuo-thermal congruency effect in the rubber hand illusion. We still had a modest RHI in the incongruent conditions (i.e., illusion composite scores >1 with positive affirmative ratings for both ownership and referral of touch statements) that seems to drive the performance at the thermal matching task. This could be the reason we did not find a main effect of congruency in the performance on the thermal matching task. Indeed, visuo-thermal incongruency can give rise to a visuo-thermal illusion when taking into account individual differences in interoceptive sensibility, whereby participants perceive the touch on their own hand as warmer or cooler than the actual temperature they feel if they look at a rubber hand touched by a visually warm or cold object, respectively. However, it remains unclear whether such a visuo-thermal illusion effect is due to the process of embodying the rubber hand or is the result of visual-thermal stimulation only (i.e., a thermal contagion-like phenomenon, that is, experiencing a specific temperature simply by looking at an ice cube or perceiving the observed temperature that someone else is experiencing, see ref. [35]). In other words, is the sense of body ownership necessary to elicit the visuo-thermal illusion effect? To answer this outstanding question, in Experiment 2, we performed a more potent manipulation to completely block the basic RHI while maintaining the same visuo-thermal stimulation used in Experiment 1. Here, we investigated the visuo-thermal illusion effect when manipulating the congruency of the observed (on the rubber hand) and felt (on the real hand) touch (see Fig. 1), while blocking the embodiment by placing the hand at the implausible position of 90° with respect to the real hand[38,39]. We included both synchronous and asynchronous conditions, in line with previous studies (e.g., ref. [14]), to further test complete RHI abolishment by showing a lack of differences between the synchronous and asynchronous anatomically incongruent conditions. Notably, if the visuo-thermal illusion effect is closely linked to the RHI, then we should not observe a visuo-thermal illusion effect in Experiment 2. In contrast, if the RHI and the visuo-thermal illusion are independent, then we might observe the visuo-thermal illusion in the direction of the observed touch, in line with previous findings obtained in the context of a more general thermal contagion phenomenon. Here, we decided to investigate the visual and tactile experience of cold perception only, since the results of Experiment 1 suggested stronger effects for cold conditions compared to warm conditions.

**Experiment 2: participants**. Thirty-three naïve participants (16 women; mean age = 26.39; SD = 5.03) were recruited using social media and advertising on the Karolinska Institutet campus. No outliers were identified (values above or below 2.5 SD from the mean in each variable). The same inclusion criteria and recruitment procedure as Experiment 1 applied (see above).

**Experiment 2: materials and experimental procedure**. Upon arrival and after signing the consent form, participants provided their demographic information and completed the BPQ questionnaire (mean BPQ = 28.94; SD = 9.29). Next, a modified version of the RHI paradigm used in Experiment 1 was performed using two custom-made panels to control visual feedback of the participants' arm and the rubber hand (which was rotated by 90°; see Fig. 6). The distance between the index fingers of the real and rubber hands was 15 cm. We had four different experimental conditions that were presented in randomised order for a total of 12 semi-randomised trials. The conditions were congruent (felt 24 °C and saw an ice cube) and incongruent (felt 32 °C and saw an ice cube), both of which could be synchronous or asynchronous. In the asynchronous conditions, there was a temporal mismatch of approximately 1500 ms between visual and tactile stimulation. The testing room was kept at a constant temperature measured by means of a room thermometer and checked at the beginning and at the end of each testing session (mean$_\text{pre}$ = 21.34, SD = 0.31; mean$_\text{post}$ = 21.86, SD = 0.28).

**Experiment 2: experimental design and statistical analysis**. Data were analysed using SPSS 26. First, we tested for normality. Proprioceptive drift was found to be non-normally distributed and, given the ordinal nature of the questionnaire data, these data were analysed using non-parametric statistics by means of the Wilcoxon signed-rank test. The limb temperature data and thermal matching task data were found to be normal, so repeated-measures ANOVAs were applied, followed by Bonferroni corrected post hoc analyses when appropriate. The same statistical strategy as Experiment 1 was followed. In addition, we compared the data from Experiments 1 and 2 by means of the Wilcoxon signed-rank test. All the reported statical tests are two-tailed.

**Statistics and reproducibility**. The details about statistics used in different data analyses performed in these studies are given in the Experimental design and statistical analysis section above. All data analyses for Experiment 1 were conducted of a total sample of 40 participants. All data analyses for Experiment 2 were conducted on a total sample of 33 participants.

**Reporting summary**. Further information on research design is available in the Nature Research Reporting Summary linked to this article.

## Data availability

The processed data that support the findings of this study are available as Supplementary Data 1. Versions of the main figures with data distribution are shown in the Supplementary Materials.

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

## Acknowledgements

We would like to thank Martti Mercurio, Bo Johansson, Zakaryah Abdulkarim for their assistance with the equipment and experimental setup. We would also like to thank Adam Enmalm for helping with data collection for Experiment 2. This work was supported by the Göran Gustafsson Foundation, the Swedish Research Council (Distinct Professor Grant) and the European Research Council under the European Union's horizon 2020 research and innovation programme (SELF-UNITY) to H. Henrik Ehrsson. Laura Crucianelli was supported by the Marie Skłodowska-Curie Intra-European Individual Fellowship (HOMEOTHERMIC SELF).

## Author contributions

L.C. and H.H.E. conceptualised and designed the study. L.C. collected and analysed the data. L.C. and H.H.E. wrote the manuscript.

## Funding

## Competing interests

The authors declare no competing interests.
