## [Peer Review File · Communications Biology]

Reviewers' comments:

Reviewer #1 (Remarks to the Author):

The study demonstrates the two-way relationship between thermal perception and body ownership. Specifically, the results show that visuo-thermal congruent stimulation of the real and rubber hand induces the embodiment of the latter. More importantly, the authors demonstrate that the embodiment of the rubber hand allows visuo-thermal incongruence to modulate thermal perception. The study has significant potential application in exploiting the two-way influence phenomenon between interception and body ownership in clinical populations.

I would like to commend the authors for the clarity and linearity of their methods. However, a few minor concerns need to be addressed.

Introduction

- The authors introduce the thermal matching task in the introduction section (page 5, from line 145; page 6, for line 174). However, the task procedure is not clear until its description in the methods section. I would suggest the authors rephrase the introduction to the thermal matching task in the introduction section.

- Page 9, line 266 (and also the discussion) refers to the illusion questionnaire. However, the questionnaire is also called 'thermo-affective' throughout the manuscript. I would ask the authors to keep a consistent nomenclature, preferably 'illusion questionnaire', as it is composed of illusion statements, the control, and thermo-affective items.

Methods (Experiments 1 and 2)

- How have the outliers been handled?
- Which test for normality has been executed? Please report the results.
- How has the handedness of the participants been investigated?

Results (Experiments 1 and 2)

- Regarding the analysis of the comparison between the subjective and objective rubber hand illusion in Experiment 1 and 2, it is not clear if the two Experiments included trials without thermal stimulation. If not, the embodiment scores of the two cannot be comparable given the different experimental conditions. All other comparisons between the two tasks must be handled carefully as the sample and conditions are different. Replicating the same paradigm of experiment one on Experiment 2's sample would have allowed for a direct comparison of some of the conditions of the two tasks.

Reviewer #2 (Remarks to the Author):

This manuscript describes two studies on the effect of visuothermal congruency on the rubber hand illusion and on a thermal perception. Visuothermal congruency was manipulated by applying stroking with a hot or a cold stimulus on the real hand and an object that is associated with hot and cold (a hand warmer and an ice cube) to the rubber hand. In experiment 1, the authors observed that using congruent visuothermal stimulation (e.g. cold stimulus on real hand, ice cube on rubber hand) resulted in a larger rubber hand illusion on both the questionnaire ratings and proprioceptive drift, than incongruent visuothermal stimulation. An effect of the congruency was also found on the thermal matching task, but only when the data on an interoceptive questionnaire was entered as covariate. A higher error in thermal matching was found in the incongruent compared to the congruent conditions especially for those who scored lower on interoceptive sensibility. To test whether this latter effect was due to the RHI or to a thermal contagion-like phenomenon, in experiment 2 a very similar procedure as in exp 1 was performed (with cold stimuli only), but now with the rubber arm rotated 90 degrees.

Indeed, no RHI was induced and while there was again a significant effect of congruency on the thermal matching errors, this time effect was reversed, i.e. higher errors in the congruent condition. The authors therefore concluded that the RHI was crucial for this visuothermal illusion. Overall, they suggest that the findings show that thermosensation contributes to the sense of body ownership, by a mechanism of dynamic integration of visual and thermosensory signals.

This is an interesting study, that fits well in a range of other studies investigating how integration of different types of sensory signals contribute to the sense of body ownership. The combination of visual and thermal features appears to be well chosen and overall, the experiments are well designed. Data analyses is generally appropriate, and interpretation of the results is correct. As such, it provides a useful addition to the current literature on multisensory mechanisms underlying body ownership. I do have a number of conceptual and methodological remarks:

Introduction

1. The classification of thermosensation as mainly (or entirely) an interoceptive sense as suggested in the Introduction and Discussion, remains somewhat controversial, in my opinion. As the authors state themselves, thermosensation also provides important information about the environment and about objects in the environment. Also, I don't think it is necessary for thermosensation to be classified as interoceptive for it to contribute to body ownership experience. I suggest leaving a discussion of thermosensation as interoceptive sense out of the Introduction, or at least describe that it contributes to both interoception and exteroception (this is mentioned in the Discussion (lines 599-600), however, in the next paragraph (p. 19, lines 601 and further), thermosensation is again considered to be an interoceptive submodality). As, in the current study, congruency is achieved by associating a certain temperature (hot or cold) with a particular object, it seems to me that the exteroceptive properties are at least as relevant as the interoceptive properties.

2. In the Introduction and Discussion, thermoregulation and thermosensation are described as being highly integrated/linked. However, there are very different concepts. One is about maintaining adequate body temperature (which involves many automatic and unconscious processes), the other is about perceiving hot and cold stimuli (which often involves a conscious experience). They are no doubt linked, but it would be important to be specific about their differences and how they relate to each other.

Methodology

3. P. 9, lines 257-260: "The task followed a staircase procedure, that is the temperature was either increased (from 24C) or decreased (from 40C) towards the reference temperature, in discrete steps of 2C. Participants were instructed to try and match the reference temperature that they previously experienced." This description is not entirely clear to me. How did the participants match the reference temperature? What kind of response or procedure did they use?

4. Analyses: In the results a non-significant interaction is reported between congruency and temperature for the non-parametric composite illusion scores (p. 11). As Wilcoxon tests only allow to test main effects, how is the non-parametric interaction performed?

Results

5. Thermal matching task (line 392): "...suggesting higher errors in the thermal matching task in the incongruent conditions as compared to congruent conditions." As these errors are directional, higher here means more positive I presume. The same is true for the results of experiment 2. Could the authors be more precise about this? Now, higher sounds like larger, but this would only apply to absolute errors.

Experiment 2

6. P. 14, lines 410-411: We still had a modest RHI in the incongruent conditions that seems to drive the performance at the thermal matching task." However, there was no significant main effect of congruency on the TMT, at least not when interoceptive sensibility is not taken into account. Please be more specific about what you mean here.

7. Experiment 2: Bayesian analyses suggests that the evidence is inconclusive with regards to the illusion questionnaire results. This was not mentioned when interpreting the results. At least, this should be mentioned as limitation in the Discussion.

Minor comments

- Figures 3, 4 and 6: Decimals are indicated using a , rather than a .
- Exp 2, line 446, p 15: "The distance between the real and rubber hand was 15 cm." I presume between the index fingers of the real and rubber hand, please be more specific in this description.
- A description of the asynchronous stimulation is missing for exp. 2. Please describe how asynchronicity was achieved.

Chris Dijkerman

Reviewer #3 (Remarks to the Author):

In this paper, the authors addressed the contribution of thermosensory information to the sense of body (upper limb) ownership. To reach this end, they performed two experiments in young healthy people using a rubber hand illusion (RHI) paradigm, which is usually performed when studying the integration of visual, vestibular, proprioceptive and tactile inputs. In the first experiment, they evaluated the impact of visuo-thermal congruency (vs incongruency), i.e temperature of the seen touch vs felt touch, on the performance in the RHI model. In a second experiment, they repeated the same paradigm with the cold temperature only but placed the rubber hand in an incongruent position (compared to the real hand) to assess if the sense of body ownership is necessary to elicit the visuo-thermal illusion effect.

I was very pleased to read this paper since it addresses an understudied factor in the construction of body representation, which is the perfect example of what multimodal integration is. If proprioceptive, vestibular or visual information are often put at the forefront, the role of temperature integration has been far less addressed. This makes this manuscript innovative and original in this field. The overall methodology is particularly sound and attractive, many factors have been taken into account and controlled, but I think some precisions could be given to better allow replicability (see below). The manuscript is well written and attractive, pleasant to read and I liked its quite concise aspect in the presentation of background, hypotheses and results.

I would like to raise several points (few of them are major ones) which will be presented following the structure of the article (and not sorted by importance).

The abstract is concise, informative and summarizes all the major points of this work

I have no specific concern about the introduction which presents a clear and progressive view of previous data and scientific gaps.

Experiment 1

Inclusion criteria (also applicable for exp 2):

- Why did the authors limit the age criteria up to 40y?
- How has the handedness been assessed?

Methodological points:

- In the methods section, the description of thermal sensors remains unprecise ("dorsal hand, dorsal forearm"). Did the authors place them following a specific methodology (measurements), and if not could this uncertainty have led to a measurement bias?
- The authors said that the real and fake hands were placed 15cm lateral to the panel. What was their landmark for the hand (medial aspect, a digit, lateral aspect)?
- Proprioceptive drift task: how was the position bias assessed (what was the precision of the measure, was the measure index-centered (index = 0) or panel centered (index = 15cm)? These elements are specified in the cited reference (Abdulkarim 2021) but I think they must be directly mentioned in the paper for a better understanding.

- The authors used a Wilcoxon test when normality assumption was not met (proprioceptive drift task). This has not allowed a comparison of each factor at the same time, as done in ANOVA. Given the sample size (n=40) and the fact that the ANOVA model is quite robust to violation of normality, why did the authors did not consider to perform an ANOVA for this dependent variable?
- P11 L308: please specify what type of correlation analyses have been performed (Spearman as stated in the suppl material).

P8 L247 to P9 L249: This point applies to the whole experiment and could be placed earlier in the text

Results:

- The order of presentation of the results (questionnaire, proprio, thermal) does not fit with the one presented in the methods section. This is a bit disturbing, why did the authors make this choice?
- P12 L360. Should "on the control score" be placed at the end of the sentence (effect of congruency and temperature on the control score)?
- Fig 3B is not mentioned in the text
- There is no mention of correlational analyses results in the text. The detail is given as supplementary data, but as it was done with skin temperature data (J540), a summarizing sentence about these results would be of interest for the reader.

Experiment 2

Methods

- Have the participants of exp 2 also participated in exp 1 or were they naïve? This could induce a bias since they already know the task and (perhaps) expected results.

Results

- L483 and 517: RHI in exp 1 and 2 has been compared using a Wilcoxon test. However, groups are not paired but independent between experiments, a Mann Withney test should have been done. In addition, can the authors specify if data of exp1 took into account warm and cold conditions or only cold ones?

Supplementary results

P7: Experiment 3 is mentioned, please correct.

Discussion

I have particularly appreciated the discussion.

Only one limitation is mentioned. I would add some concerning statistical analyses (particularly concerning Wilcoxon tests as mentioned above).

Replies to Reviewers' comments

Reviewer #1 (Remarks to the Author):

The study demonstrates the two-way relationship between thermal perception and body ownership. Specifically, the results show that visuo-thermal congruent stimulation of the real and rubber hand induces the embodiment of the latter. More importantly, the authors demonstrate that the embodiment of the rubber hand allows visuo-thermal incongruence to modulate thermal perception. The study has significant potential application in exploiting the two-way influence phenomenon between interception and body ownership in clinical populations. I would like to commend the authors for the clarity and linearity of their methods. However, a few minor concerns need to be addressed.

Response: We would like to thank the Reviewer for their time and for the positive evaluation of the manuscript. We are happy that the Reviewer appreciated the potential of our study, and we are most grateful for their comments that have helped to improve the clarity of the manuscript. We have now addressed their concerns in turn below.

Introduction

POINT 1: The authors introduce the thermal matching task in the introduction section (page 5, from line 145; page 6, for line 174). However, the task procedure is not clear until its description in the methods section. I would suggest the authors rephrase the introduction to the thermal matching task in the introduction section.

Response: We thank the Reviewer for this comment. We have now edited the manuscript accordingly, so that we provide more details about the thermal matching task in the introduction. Please see below and page 5 of the revised manuscript:

Moreover, to investigate a possible visuo-thermal illusion effect, that is, that the visual impressions of the object touching the rubber hand would influence the perceived temperature of the object in incongruent conditions but only when participants experienced some degree of rubber hand illusion; not when it was eliminated, participants completed a thermal matching task (Crucianelli, Enmalm & Ehrsson, 2021). In its original format, participants were stroked with a thermode attached to a thermal stimulator at a 'reference temperature'. The task of the participants is to match the reference temperature of the touch when presented among an ascending or descending series of successfully warmer (up to +8 °C) or cooler (up to -8 °C) temperatures in a staircase procedure. Here, participants completed the thermal matching task following the rubber hand illusion, whereby the reference temperature was always the temperature felt during the induction of the illusion, thus providing an objective measure of thermosensation (see Methods for more details).

POINT 2: Page 9, line 266 (and also the discussion) refers to the illusion questionnaire. However, the questionnaire is also called 'thermo-affective' throughout the manuscript. I would ask the authors to keep a consistent nomenclature, preferably 'illusion questionnaire', as it is composed of illusion statements, the control, and thermo-affective items.

Response: We apologise for the lack of clarity from our side. We have now edited the entire manuscript throughout to increase consistency. The Reviewer is right; indeed, the ‘illusion questionnaire’ is composed of of illusion statements, the control, and thermo-affective items. We now refer to it as the (rubber hand) illusion questionnaire.

Methods (Experiments 1 and 2)

POINT 3: How have the outliers been handled?

Response: We have not identified any outliers in both experiments; hence we have not included information about this on the manuscript. We have followed the conventional rule of defining as outliers values above or below 2.5 SD from the mean for each variable. We have now added this information to the manuscript. Please see page 7 and 14-15 of the revised version:

No outliers were identified (values above or below 2.5 SD from the mean in each variable).

POINT 4: Which test for normality has been executed? Please report the results.

Response: Normality has been tested by means of Shapiro-Wilk tests and we checked the data distribution also by means of visual exploration (by looking at Q-Q-Plots). We have now included more details in the text (page 10). Please see below:

Data were analysed using SPSS 26. First, we tested for normality by means of Shapiro–Wilk tests, and data distribution was also checked by means of visual exploration (Q-Q-Plots). Proprioceptive drift was found to be non-normally distributed (p values < 0.05), and given the ordinal nature of the questionnaire data, these data were analysed using non-parametric statistics by means of Wilcoxon signed-ranks test.

POINT 5: How has the handedness of the participants been investigated?

Response: Handedness has been investigated via self-report, by recruiting only participants who considered themselves as being fully right-handed. We have not performed a formal assessment.

Results (Experiments 1 and 2)

POINT 6: Regarding the analysis of the comparison between the subjective and objective rubber hand illusion in Experiment 1 and 2, it is not clear if the two Experiments included trials without thermal stimulation. If not, the embodiment scores of the two cannot be comparable given the different experimental conditions. All other comparisons between the two tasks must be handled carefully as the sample and conditions are different. Replicating the same paradigm of experiment one on Experiment 2’s sample would have allowed for a direct comparison of some of the conditions of the two tasks.

Response: We agree with the Reviewer here that the comparison between the two experiment is not ideal. We would like to clarify that the main purpose for conducting Experiment 2 was not to have a direct comparison with Experiment 1, but it is rather a control experiment to check for the effect of body ownership on visuo-thermal illusion. Thus, we agree that the direct comparison between the two

experiment is not optimal and was beyond the scope of this paper (indeed we defined it as exploratory). Nevertheless, we believe it is informative, and it gives a sense of the effects of interest, therefore we have included it in supplementary. As suggested by the Reviewer, we have indeed compared only the conditions where the thermal stimulation was identical between Experiment 1 and 2, and we made this even more clear in the manuscript. We hope that future experiments will replicate and extend our findings. Please see page 7 of the revised supplementary materials and below:

In an exploratory analysis, we combined the data from Experiment 1 and Experiment 2 for those conditions where the thermal stimulation was identical across the two experiments (see Figure 1 in the main manuscript). In particular, we were interested in comparing the congruent cold condition (i.e., participant feeling the touch at 24°C on their skin while looking at the rubber hand being touched by an ice cube) across Experiment 2 (classic rubber hand illusion) and Experiment 3 (hand placed at 90°).

Reviewer #2 (Remarks to the Author):

This manuscript describes two studies on the effect of visuothermal congruency on the rubber hand illusion and on a thermal perception. Visuothermal congruency was manipulated by applying stroking with a hot or a cold stimulus on the real hand and an object that is associated with hot and cold (a hand warmer and an ice cube) to the rubber hand. In experiment 1, the authors observed that using congruent visuothermal stimulation (e.g. cold stimulus on real hand, ice cube on rubber hand) resulted in a larger rubber hand illusion on both the questionnaire ratings and proprioceptive drift, than incongruent visuothermal stimulation. An effect of the congruency was also found on the thermal matching task, but only when the data on an interoceptive questionnaire was entered as covariate. A higher error in thermal matching was found in the incongruent compared to the congruent conditions especially for those who scored lower on interoceptive sensibility. To test whether this latter effect was due to the RHI or to a thermal contagion-like phenomenon, in experiment 2 a very similar procedure as in exp 1 was performed (with cold stimuli only), but now with the rubber arm rotated 90 degrees. Indeed, no RHI was induced and while there was again a significant effect of congruency on the thermal matching errors, this time effect was reversed, i.e. higher errors in the congruent condition. The authors therefore concluded that the RHI was crucial for this visuothermal illusion. Overall, they suggest that the findings show that thermosensation contributes to the sense of body ownership, by a mechanism of dynamic integration of visual and thermosensory signals.

This is an interesting study, that fits well in a range of other studies investigating how integration of different types of sensory signals contribute to the sense of body ownership. The combination of visual and thermal features appears to be well chosen and overall, the experiments are well designed. Data analyses is generally appropriate, and interpretation of the results is correct. As such, it provides a useful addition to the current literature on multisensory mechanisms underlying body ownership. I do have a number of conceptual and methodological remarks:

Response: We would like to thank the Reviewer for their time and for the positive evaluation of the manuscript. We are happy that the Reviewer appreciated the potential of our study, and we are most grateful for their comments that have helped to improve the quality of the manuscript. We have now addressed their concerns in turn below.

Introduction

POINT 1: The classification of thermosensation as mainly (or entirely) an interoceptive sense as suggested in the Introduction and Discussion, remains somewhat controversial, in my opinion. As the authors state themselves, thermosensation also provides important information about the environment and about objects in the environment. Also, I don't think it is necessary for thermosensation to be classified as interoceptive for it to contribute to body ownership experience. I suggest leaving a discussion of thermosensation as interoceptive sense out of the Introduction, or at least describe that it contributes to both interoception and exteroception (this is mentioned in the Discussion (lines 599-600), however, in the next paragraph (p. 19, lines 601 and further), thermosensation is again considered to be an interoceptive submodality). As, in the current study, congruency is achieved by associating a certain temperature (hot or cold) with a particular object, it seems to me that the exteroceptive properties are at least as relevant as the interoceptive properties.

Response: We thank the Reviewer for this comment that allows us to clarify further the rationale of our study. We completely agree that thermosensation provides both interoceptive and exteroceptive signals, and we also agree regarding the fact that there is still debate on whether skin-mediated signals should be considered as interoceptive overall. We have recently published a paper at this regard, where we extensively unwrapped the potential of thermosensation as an interoceptive modality, taking also into consideration the exteroceptive component (Crucianelli & Ehrsson, 2022). We agree that the exteroceptive properties of perceived temperature are as relevant as the interoceptive ones when one explores objects; however, the rationale behind the current study was to take advantage of the fact that thermosensation has a clear interoceptive dimension (for example, the signals from c-fibers transmitted through a special anatomical pathway through the spinal cord via the thalamus to the insular cortex compared to discriminative touch; and many other arguments see Crucianelli & Ehrsson 2022) and that it can be considered as an interoceptive modality and not only an exteroceptive one that has been the traditional view for most of the 20th century. Thus thermosensation provide both interoceptive and exteroceptive information even when objects touch the skin. In order to emphasize the interoceptive aspect in the current study, we have tried to keep the exteroceptive components of touch constant (e.g., the object delivery the touch was always the same, velocity and frequency of touch were within the CT-optimal range, constant pressure, etc.) while manipulating only the thermal signal, which supposedly provide interoceptive information. Thus here, we are not interested in looking at temperature as another aspect of exteroceptive somatosensation, but rather as skin-mediated interoceptive somatosensory signals. Thus, we believe that it is important to keep the notion of thermosensation as an interoceptive modality in the introduction too, because this is very much part of the rationale of the study and subsequent experimental choices. Nevertheless, we have stressed this duality of thermosensation even more throughout the manuscript. We have now included our recent preprint as we think it could provide an important insight into this debate.

Crucianelli, L., & Ehrsson, H. H. (2022). The role of the skin in interoception: A neglected organ? PsyArXiv, 10.31234/osf.io/qfu87

Please see below and page 3 of the revised manuscript:

Thermal information is conveyed from the skin via thin unmyelinated c-fibres and takes a separate pathway to the brain compared to that of discriminative touch that travels through the spinal cord

(spinal lamina I) and thalamus (ventral medial posterior nucleus) and reaches the insular cortex; the insular cortex is an important cortical region for processing interoceptive signals (including information from visceral organs). These are some of the reasons temperature and the affective component of touch, among others, have been redefined as interoceptive modalities (Craig, 2002; Crucianelli et al., 2018; Crucianelli, Enmalm & Ehrsson, 2021) and separated from exteroceptive sensations (e.g., Ferentzi, Bogdány, Szabolcs, Csala, Horváth & Köteles, 2018), such as information about the external environment (e.g., visual stimuli or discriminatory touch) and proprioception, which provides sensations of the positions of limbs and body parts in space (Walsh, Moseley, Taylor & Gandevia, 2011). Interoception refers to the perception and representation of internal signals about the physiological status of the body (Craig, 2002). Thermosensation both provides information about the thermal state of one's own body (interoception) and about the thermal properties of the environment (exteroception), and, therefore, it can be used as an attractive model system of skin-based interoception (Crucianelli & Ehrsson 2022). However, despite the tight link among thermosensation, interoception, and bodily awareness, there has been hardly any investigation of the relationship among these three aspects (Crucianelli, Enmalm & Ehrsson, 2021; Crucianelli & Ehrsson, 2022).

POINT 2: In the Introduction and Discussion, thermoregulation and thermosensation are described as being highly integrated/linked. However, there are very different concepts. One is about maintaining adequate body temperature (which involves many automatic and unconscious processes), the other is about perceiving hot and cold stimuli (which often involves a conscious experience). They are no doubt linked, but it would be important to be specific about their differences and how they relate to each other.

Response: This is an interesting and valid point indeed. We agree with the Reviewer that thermoregulation and thermosensation are two distinct phenomena albeit highly related. We have now included a paragraph in the discussion to elaborate on the links and differences between these two concepts. Please see below and page 4 of the revised manuscript:

Furthermore, the studies conducted thus far have mainly focused on investigating the link between body ownership and possible changes in skin temperature (thermoregulation, e.g., Moseley et al., 2008) rather than thermosensation. Although highly linked to one another, thermoregulation and thermosensation are two distinct phenomena, and they both contribute to the maintenance of thermoneutrality. Thermoregulation mainly involves automatic processes, while thermosensation is related to the conscious perception of thermal stimuli via the skin (e.g., Kurz, 2008; Filingeri, Zhang & Arens, 2018; Morrison & Nakamura, 2019).

Filingeri, D., Zhang, H., & Arens, E. A. (2018). Thermosensory micromapping of warm and cold sensitivity across glabrous and hairy skin of male and female hands and feet. *Journal of Applied Physiology*, 125(3), 723-736.

Kurz, A. (2008). Physiology of thermoregulation. *Best Practice & Research Clinical Anaesthesiology*, 22(4), 627-644.

We also discussed this issue in the discussion (see page 22):

Moreover, these findings also suggest that visuo-thermal mismatches between the felt and seen touch in the RHI do not significantly change the skin temperature to indicate a physiological homeostatic thermal feedback response (as in Cooper et al., 2014), at least not significantly with the set of stimuli used in the current study (see supplementary material, Experiment 1). This result is also in line with the view that thermosensation and thermoregulation involve different mechanisms and that they can deviate.

Methodology

POINT 3: P. 9, lines 257-260: “The task followed a staircase procedure, that is the temperature was either increased (from 24C) or decreased (from 40C) towards the reference temperature, in discrete steps of 2C. Participants were instructed to try and match the reference temperature that they previously experienced.” This description is not entirely clear to me. How did the participants match the reference temperature? What kind of response or procedure did they use?

Response: We apologise for any lack of clarity from our side. We are happy to provide more details about the task. We have included more details on the introduction, as suggested by Reviewer 1 and on the method. We also hope that interested reader would refer back to the original paper describing the *thermal matching task* for reference (Crucianelli, Enmalm & Ehrsson, 2021). Please see below and page 9 of the revised manuscript:

Participants were instructed to try and match the reference temperature that they previously experienced by verbally saying ‘stop’ when the test temperature presented among warmer or cooler temperatures in the staircase procedure matched the reference temperature. In the post-trial thermal matching task, we repeated the same procedure, but participants were asked to match the temperature they had felt on their real hand while they were looking at the rubber hand (reference temperature) by verbally saying ‘stop’ when the series of warmer and cooler temperature was presented (after the illusion induction stimulation).

See also page 5 of the introduction:

Moreover, to investigate a possible visuo-thermal illusion effect, that is that the visual impressions of the object touching the rubber hand would influence the perceived temperature of the object in the incongruent conditions, but only when participants experienced some degree of rubber hand illusion and not when it was eliminated, participants completed a thermal matching task (Crucianelli, Enmalm & Ehrsson, 2021). In its original format, participants are stroked with a thermode attached to a thermal stimulator at a ‘reference temperature’. The task of the participants is to match the reference temperature of the touch, when this is presented among an ascending or descending series of successfully warmer (up to +8°C) or cooler (up to -8°C) temperatures in a staircase procedure. Here,

participants completed the thermal matching task following the rubber hand illusion, whereby the reference temperature was always the temperature felt during the induction of the illusion, thus providing an objective measure of thermosensation (see Methods for more details).

POINT 4: Analyses: In the results a non-significant interaction is reported between congruency and temperature for the non-parametric composite illusion scores (p. 11). As Wilcoxon tests only allow to test main effects, how is the non-parametric interaction performed?

Response: We apologise for the lack of clarity from our side. We now provide more details on how the interaction was performed in section 2.3. *Experimental design and statistical analysis*. We used the same procedure used in Crucianelli et al 2013. Please see below and page 10 of the revised manuscript:

The interaction between congruency and temperature was analysed by calculating the difference between congruent and incongruent scores in the warm and cold conditions separately; subsequently, a Wilcoxon signed rank test was performed to compare these two difference scores.

Results

POINT 5: Thermal matching task (line 392): "...suggesting higher errors in the thermal matching task in the incongruent conditions as compared to congruent conditions." As these errors are directional, higher here means more positive I presume. The same is true for the results of experiment 2. Could the authors be more precise about this? Now, higher sounds like larger, but this would only apply to absolute errors.

Response: This is an important point indeed, thank you for noticing it. We did not use absolute values; therefore, the errors are indeed directional. That is why, in Figure 3 for example, we have negative values for the TMT errors.

Experiment 2

POINT 6: P. 14, lines 410-411: We still had a modest RHI in the incongruent conditions that seems to drive the performance at the thermal matching task." However, there was no significant main effect of congruency on the TMT, at least not when interoceptive sensibility is not taken into account. Please be more specific about what you mean here.

Response: We agree that perhaps this was not very clear, apologies. What we meant is that maybe we did not find a significant main effect of congruency on the TMT because we still had affirmative ownership and referral of touch ratings in the incongruent condition (but significantly lower so compared to the congruent conditions). In other words, the incongruent condition significantly reduced the illusion but did not fully abolish it, and the composite illusion score was still above the score 1. Thus, we needed to run a second experiment in which we eliminated the RHI (clearly negative illusion ratings) to be able to disentangle the role of body ownership in the visuo-thermal illusion. Please see below and page 14 of the revised manuscript:

The results of Experiment 1 showed a visuo-thermal congruency effect in the rubber hand illusion. We still had a modest RHI in the incongruent conditions (i.e., illusion composite scores > 1 with positive affirmative ratings for both ownership and referral of touch statements) that seems to drive the

performance at the thermal matching task. This could be the reason we did not find a main effect of congruency in the performance on the thermal matching task.

POINT 7: Experiment 2: Bayesian analyses suggests that the evidence is inconclusive with regards to the illusion questionnaire results. This was not mentioned when interpreting the results. At least, this should be mentioned as limitation in the Discussion.

Response: We think this could be a misunderstanding because the questionnaire results clearly show that the rubber hand illusion was denied by most participants in Experiment 2 (mean negative ratings lower than -1 in all conditions, see Figure S1) in line with the well-established abolishing effect of the anatomically impossible 270 degrees counter-clockwise postural manipulation we have used (Tsakiris et al 2005; Ide, 2013; see also results from 180 degrees manipulation e.g., Ehrsson et al., 2004). Furthermore, Experiment 2's orientation manipulation led to significantly weaker rubber hand illusion ratings than Experiment 1 when we statistically compared the ratings across experiments. So, the questionnaire results in Experiment 2 were not inconclusive with respect to rubber hand illusion or the experimental manipulation. The inconclusive Bayesian results the reviewer mentioned only refer to the non-significant differences between the synchronous and the asynchronous condition when the rubber hand was placed in the anatomically incongruent conditions. So overall, we do not think that the inclusive Bayesian evidence in support of the null hypothesis in this regard is a major limitation of the study that needs to be considered in the discussion paragraph. But we have edited a sentence in the Results section so that we now better contextualize this result to avoid any misunderstanding. Please see page 18 of the revised manuscript and below:

Thus, the rubber hand illusion was denied by most participants in both synchronous and asynchronous conditions (mean negative ratings < -1 in all conditions, see Figure S1).

Minor comments

POINT 7: Figures 3, 4 and 6: Decimals are indicated using a , rather than a .

Response: We thank the Reviewer for pointing this out. We have now edited the figures accordingly.

POINT 8: Exp 2, line 446, p 15: "The distance between the real and rubber hand was 15 cm." I presume between the index fingers of the real and rubber hand, please be more specific in this description.

Response: Yes, the Reviewer is right. We took as reference point the distance between the index fingers of the real and rubber hand. We have included this detail in the manuscript as follows (see also page 15 of the revised manuscript):

The distance between the index fingers of real and rubber hand was 15 cm.

POINT 9: A description of the asynchronous stimulation is missing for exp. 2. Please describe how asynchronicity was achieved.

Response: We have added a sentence in the manuscript to better clarify how the asynchronous condition was achieved. Please see below and page 15 of the revised manuscript:

In the asynchronous conditions, there was a temporal mismatch of approximately 1500 ms between visual and tactile stimulation.

Chris Dijkerman

Reviewer #3 (Remarks to the Author):

In this paper, the authors addressed the contribution of thermosensory information to the sense of body (upper limb) ownership. To reach this end, they performed two experiments in young healthy people using a rubber hand illusion (RHI) paradigm, which is usually performed when studying the integration of visual, vestibular, proprioceptive and tactile inputs. In the first experiment, they evaluated the impact of visuo-thermal congruency (vs incongruency), i.e temperature of the seen touch vs felt touch, on the performance in the RHI model. In a second experiment, they repeated the same paradigm with the cold temperature only but placed the rubber hand in an incongruent position (compared to the real hand) to assess if the sense of body ownership is necessary to elicit the visuo-thermal illusion effect.

I was very pleased to read this paper since it addresses an understudied factor in the construction of body representation, which is the perfect example of what multimodal integration is. If proprioceptive, vestibular or visual information are often put at the forefront, the role of temperature integration has been far less addressed. This makes this manuscript innovative and original in this field. The overall methodology is particularly sound and attractive, many factors have been taken into account and controlled, but I think some precisions could be given to better allow replicability (see below). The manuscript is well written and attractive, pleasant to read and I liked its quite concise aspect in the presentation of background, hypotheses and results.

I would like to raise several points (few of them are major ones) which will be presented following the structure of the article (and not sorted by importance).

The abstract is concise, informative and summarize all the major points of this work

I have no specific concern about the introduction which presents a clear and progressive view of previous data and scientific gaps.

Response: We would like to thank the Reviewer for their time and for the positive evaluation of the manuscript. We are happy that the Reviewer appreciated the aims of our study, and we are most grateful for their careful comments that have helped to improve the manuscript. We have now addressed their concerns in turn below.

Experiment 1

POINT 1: Inclusion criteria (also applicable for exp 2):

- Why did the authors limit the age criteria up to 40y?
- How has the handedness been assessed?

Response: The effect of age on the rubber hand illusion remains unclear, as the results are so far inconsistent (see for example, Palomo, Borrego, Cebolla, Llorens, Demarzo, & Baños, 2018; Ferracci and Brancucci, 2019; Marotta, Zampini, Tinazzi, & Fiorio, 2018). In particular, a few studies showed that the susceptibility to the RHI might decrease from middle-adulthood (>40 years old), therefore we decided to keep the sample consistent within the same age range. We also know that multisensory

integration changes over time with older individuals (>40 years old) showing somewhat different results compared to younger. The age range between 20 and 35 years old is usually considered as early adulthood, as defined by Palomo et al, 2018. We were not interested into looking for an effect of age, so we thought it made sense to recruit participants within the same age group.

In terms of handedness, this was investigated via self-report, by recruiting only participants who considered themselves as fully right-handed. We have not performed a formal assessment

POINT 2: Methodological points:

- In the methods section, the description of thermal sensors remains unprecise (“dorsal hand, dorsal forearm”). Did the authors place them following a specific methodology (measurements), and if not could this uncertainty have led to a measurement bias?

Response: We apologise for the lack of details from our side. We placed the thermal sensors always on the same location, which was carefully matched with the one of the fake sensors placed on the rubber hand, as shown on Figure 2 and Figure 5 of the main manuscript.

Since the data of the skin temperature have been used to compare participants’ measurements between conditions and within participants, we believe that the presence of a measurement bias is highly unlikely and should not be an issue (the sensors were placed on the same position on the skin from the beginning to the end of the experiment, so the position of the sensors was the same across participants). Furthermore, it is common in the field to say that the temperature was recorded from the middle of the dorsal hand and middle of dorsal forearm (for example see de Haan, A. M., Van Stralen, H. E., Smit, M., Keizer, A., Van der Stigchel, S., & Dijkerman, H. C. (2017). No consistent cooling of the real hand in the rubber hand illusion. *Acta psychologica*, 179, 68-77). Please see below and page 8 of the revised manuscript:

Before starting the experimental procedure, three thermal sensors (Biopac MP150) were placed on the participants’ skin to continuously measure the temperature of the skin; one sensor was placed on the right dorsal hand, one was placed on the left dorsal hand and one was placed on the left dorsal forearm (see Figure 5 for a clear view of the sensor positions).

POINT 3: The authors said that the real and fake hands were placed 15cm lateral to the panel. What was their landmark for the hand (medial aspect, a digit, lateral aspect)?

Response: We took as reference point the distance between the index fingers of the real and rubber hand. We have included this detail in the manuscript as follows (see also page 15 of the revised manuscript):

The distance between the index fingers of real and rubber hand was 15 cm.

POINT 4: Proprioceptive drift task: how was the position bias assessed (what was the precision of the measure, was the measure index-centered (index = 0) or panel centered (index = 15cm)? These elements are specified in the cited reference (Abdulkarim 2021) but I think they must be directly mentioned in the paper for a better understanding.

Response: We asked participants to indicate the position of their left index finger before and after the induction of the illusion. Namely, the experimenter asked the participant to close their eyes and placed

the participant's right index finger on the right side of the metal ruler (the starting point on the metal ruler was varied randomly by the experimenter). We then calculated the difference in cm between the actual location of the index finger and the value indicated by the participant. The proprioceptive drift was then calculated as the difference between the localization error obtained before and after the illusion. Therefore, the reference point was always the index finger. However, we believe that it is not important whether the measure is index-centered or panel-centered, since the measure of interest is always calculated as the difference between pre and post. We have now included further details in the text. Please see below and page 8-9 of the revised manuscript:

During the proprioceptive drift trials, participants were asked to close their eyes, and the experimenter positioned the participants' right index fingers on the right side of a metal ruler, which was placed on the table five centimetres over both the right hand and the rubber hand. The starting point on the metal ruler was varied randomly by the experimenter. The custom-made divider had a window in the middle that was open during the proprioceptive drift trials, allowing the participants to reach the left-hand side of the panel (see Figure 2B, see Abdulkarim, Hayatou, & Ehrsson, 2021 for a similar procedure). Then, the participants moved their right index fingers to signal where they felt that their left index finger was located. The experimenter noted the value and calculated the difference in cm between the actual location of the index finger and the one indicated by the participants to obtain a measure of localization error. The same procedure was repeated before and after each stimulation period. The proprioceptive drift was then calculated as the difference between the localization errors before and after the illusion trials.

POINT 5: The authors used a Wilcoxon test when normality assumption was not met (proprioceptive drift task). This has not allowed a comparison of each factor at the same time, as done in ANOVArm. Given the sample size ($n=40$) and the fact that the ANOVA model is quite robust to violation of normality, why did the authors did not consider to perform an ANOVArm for this dependent variable?

Response: The proprioceptive drift data did not meet normality assumptions so that is way we used non-parametric tests. But we know parametric tests have often been used in the previous literature and that proprioceptive drift data often meet normality assumptions. Therefore and in response to the reviewer's comment we have now run the analysis for the proprioceptive drift task using a parametric ANOVA too, and the results follow the same pattern of results (main effect of congruency: $F(1, 39) = 6.545, p = 0.015$; no significant main effect of temperature: $F(1, 39) = 0.135, p = 0.715$, and non-significant interaction between congruency and temperature: $F(1, 39) = 2.343, p = 0.134$. We have now added a sentence in the results section to increase the confidence in our results. Please see below and page 11 of the revised manuscript:

For confirmation purposes, we performed the same analysis using a repeated-measures ANOVA and found the same pattern of results (main effect of congruency: $F(1, 39) = 6.545, p = 0.015$; no significant main effect of temperature: $F(1, 39) = 0.135, p = 0.715$, and non-significant interaction between congruency and temperature: $F(1, 39) = 2.343, p = 0.134$).

POINT 6: P11 L308: please specify what type of correlation analyses have been performed (Spearman as stated in the suppl material).

Response: Yes, the Reviewer is right. We have now added this information on page 11. Please see below:

The illusion shift, proprioceptive shift, and thermal shift were then used in Spearman correlational analyses to investigate the relationship between the outcome measures of the RHI and the interoceptive sensibility measure. The results of these analyses are reported in the Supplementary Materials.

POINT 7: P8 L247 to P9 L249: This point applies to the whole experiment and could be placed earlier in the text

Response: We have now moved this sentence earlier in the text. Please see below and 7 of the revised manuscript:

The testing room was kept at a constant temperature measured by means of a room thermometer that was checked at the beginning and at the end of each testing session ($Mean_{pre} = 21.48\text{ }^{\circ}\text{C}$, $SD = 0.35$; $Mean_{post} = 21.76\text{ }^{\circ}\text{C}$, $SD = 0.40$).

Results:

POINT 8: The order of presentation of the results (questionnaire, proprio, thermal) does not fit with the one presented in the methods section. This is a bit disturbing, why did the authors make this choice?

Response: We apologise for this; the Reviewer is right. We have now changed the order of the results as to fit the one of the methods, that is: proprioceptive drift, thermal matching task and illusion questionnaire. This is also the order we followed in the experimental procedure, so it makes sense to have it as such. We have also changed the number of the figures accordingly. We did the same for Experiment 2.

POINT 9: P12 L360. Should “on the control score” be placed at the end of the sentence (effect of congruency and temperature on the control score)?

Response: We agree with the Reviewer that it reads better with “on the control score” at the end of the sentence, and it is more clear. We have now edited the sentence accordingly (see page 14):

The results of the Wilcoxon signed ranks test revealed a non-significant main effect of congruency ($Z = -1.00$, $p = 0.32$, $mean\ congruent = -0.98$, $SD = 1.07$; $mean\ incongruent = -1.07$, $SD = 1.15$) or temperature ($Z = -0.82$, $p = 0.42$, $mean\ cold = -1.08$, $SD = 1.09$; $mean\ warm = -1.01$, $SD = 1.14$) on the control score.

POINT 10: Fig 3B is not mentioned in the text

Response: We thank the Reviewer for pointing this out. We have now added the mention to Figure 3B in the text.

POINT 11: There is no mention of correlational analyses results in the text. The detail is given as supplementary data, but as it was done with skin temperature data (J540), a summarizing sentence about these results would be of interest for the reader.

Response: We have now added a sentence to mention the direction of the results of the correlational analyses. Please see below and page 18 of the revised manuscript:

The non-significant results of the correlational analyses across illusion measures are reported in Supplementary Materials.

Experiment 2

Methods

POINT 12: Have the participants of exp 2 also participated in exp 1 or were they naïve? This could induce a bias since they already know the task and (perhaps) expected results.

Response: We recruited different participants for Experiment 1 and 2, so the participants of Experiment 2 were naïve. We have further clarified this in the manuscript. Please see below and page 14 of the revised manuscript:

Thirty-three naïve participants (16 women; Mean age = 26.39; SD = 5.03) were recruited using social media and advertising in the Karolinska Institutet campus. The same inclusion criteria and recruitment procedure as Experiment 1 apply (see above).

Results

POINT 13: L483 and 517: RHI in exp 1 and 2 has been compared using a Wilcoxon test. However, groups are not paired but independent between experiments, a Mann Withney test should have been done. In addition, can the authors specify if data of exp1 took into account warm and cold conditions or only cold ones?

Response: We thank the Reviewer for pointing this out. We have now re-run the analysis comparing data from Experiment 1 and 2 using a Mann-Withney test instead. Please see below and page 16-17 of the revised manuscript:

We performed an additional comparison between Experiment 1 and Experiment 2 to test the hypothesis of a weaker illusion in the latter due to anatomical incongruence; the results of the Mann–Whitney test revealed a significant main effect of experiment on proprioceptive drift ($U = -4.00, p < 0.01$), with participants in Experiment 1 showing higher values of proprioceptive drift of the left hand towards the rubber hand compared to that observed in Experiment 2 ($M_{Exp 1} = 1.92, SD = 2.58; M_{Exp 2} = -0.88, SD = 3.26$).

Supplementary results

POINT 14: P7: Experiment 3 is mentioned, please correct.

Response: Thank you for pointing out this typo. We have now corrected it.

Discussion

POINT 15: I have particularly appreciated the discussion.

Only one limitation is mentioned. I would add some concerning statistical analyses (particularly concerning Wilcoxon tests as mentioned above).

Response: We are glad to hear that the Reviewer has appreciated the discussion, thank you. We politely disagree regarding the fact that the use of Wilcoxon tests should be included as a limitation of the study. We believe that when the data are non-normally distributed, non-parametric analyses is a valuable option. We have now added more details about the methods used to run the interactions with Wilcoxon tests that might increase the confidence in our results. Furthermore, since we replicate the same results with parametric tests, we believe that this is a further confirmation of the validity of our Wilcoxon tests.

REVIEWERS' COMMENTS:

Reviewer #1 (Remarks to the Author):

I thank the authors for addressing all the reviewer's points. I am happy to recommend the manuscript publication in Communications Biology.

Reviewer #2 (Remarks to the Author):

The authors have satisfactorily addressed almost all my previous comments. Only one minor issue remains:

Figures 3, 4 and 6 still contain , instead of . for the decimals

Reviewer #3 (Remarks to the Author):

I thank the authors for their work in this revised version of the manuscript.
All the concerns I have raised have been addressed, I have no additional comment.